# Self-Improvement as Coherence Optimization:
# A Theoretical Account

**Tianyi Qiu**[*]                                                                                    *tianyiq@stanford.edu*
*University of Oxford*
*Stanford University*

**Ahmed Hani Ismail**                                                                        *ahmedhismail@berkeley.edu*
*UC Berkeley*

**Zhonghao He**                                                                        *hezhonghao2030@gmail.com*
*University of Oxford*

**Shi Feng**                                                                                            *shi.feng@gwu.edu*
*George Washington University*

**Reviewed on OpenReview:** `https://openreview.net/forum?id=nR47qAX9oL`

## Abstract

Can language models improve their accuracy without external supervision? Methods such as debate, bootstrap, and internal coherence maximization achieve this surprising feat, even matching golden finetuning performance. Yet why they work remains theoretically unclear. We show that they can all be understood as *coherence optimization*, the search for a context-to-behavior mapping that is most compressible and jointly predictable, with debate an exact instance and bootstrap and internal coherence maximization closely related to it. We prove that coherence optimization is equivalent to description-length regularization, and that among all such regularization schemes, coherence regularization with a prior derived from a pretrained model optimizes a lower bound of worst-case accuracy for semi-supervised learning (Theorem 5.5). Our theory, supported by preliminary experiments, explains why feedback-free self-improvement works and predicts when it should succeed or fail.[1]

## 1 Introduction

Language models can improve their accuracy without external supervision. Methods such as debate (Irving et al., 2018; Khan et al., 2024), internal coherence maximization (ICM) (Wen et al., 2025),[2] iterative bootstrap (Lee et al., 2013; Xu et al., 2023), and Metropolis-Hastings sampling (Karan & Du, 2025) have matched supervised finetuning performance on ground-truth labels, despite using nothing but unsupervised questions and a pretrained model not finetuned on the present task. This is surprising. Where does the "supervision" come from? As we will show, the supervised signal is already embedded in the pretrained prior. The coherence optimization process itself is unsupervised, but viewed as part of the full training pipeline, the overall method is semi-supervised.

This paper provides a theoretical answer. We show that these methods optimize, exactly or approximately, the same objective, which we call *coherence*. Coherence is the joint likelihood of the model's behaviors across

---

[*]Correspondence to Tianyi Qiu `<tianyiq@stanford.edu>`.

[1]Specifically, Theorem 5.5 shows that self-improvement succeeds when the prior is a good approximation to the data-generating distribution, and Conjecture 5.9 identifies when purely unsupervised coherence optimization recovers the regularized training objective.

[2]Connections and differences between ICM and coherence optimization is discussed in §2.

all contexts, with the joint likelihood in turn defined through autoregressive conditioning or training. We prove that coherence optimization is equivalent to description-length regularization, and that among such schemes, coherence regularization with a prior derived from a pretrained model optimizes a lower bound of worst-case accuracy for semi-supervised learning.

**The Problem (Semi-Supervised Learning).** Consider a model responding to different contexts (e.g., question prompts), where each context is associated with a large space of possible behaviors (e.g., free-form answers). A *deterministic policy* (d-policy) is a complete assignment of one behavior to each context. Given $N$ supervised labels, we want to find the d-policy that generalizes best to unseen contexts. The classical solution is empirical risk minimization (ERM), which fits the supervised labels exactly, but ERM overfits when the hypothesis space is large, as is the case for language models (Lampinen et al., 2025). By utilizing access to unlabeled contexts, we would like to improve the generalizability of the learned hypothesis.

**The Solution (Coherence as Regularization).** Statistical learning theory addresses overfitting through regularization that penalizes complex hypotheses (Vapnik, 1999). Description-length regularizers, which penalize hypotheses that are hard to compress under a prior distribution, form a classical and general family within this framework. The most famous such prior is the Solomonoff prior (Solomonoff, 2009). We prove that among all description-length regularizers for semi-supervised learning, the one that optimizes the worst-case lower bound on accuracy uses the prior that best approximates the data-generating distribution (Theorem 5.5). For language models, this is the pretrained model. The description length under this prior equals the negated coherence. Our prior-comparison experiments are consistent with this conclusion, though we do not run a head-to-head comparison against other regularizers.

Formally speaking, we prove that among all description-length regularization schemes for semi-supervised learning, coherence regularization derived from a KL-optimal prior optimizes a lower bound of worst-case accuracy (Theorem 5.5). Since the optimal prior is the one that minimizes KL divergence to the data-generating distribution, and a pretrained language model is the best practically obtainable such approximation (having been trained on the largest available sample), pretrained models give the tightest such worst-case guarantee among available priors. We discuss the limits of this kind of optimality claim in §5.2.

**The Algorithm (Gibbs Sampling).** Direct optimization of coherence is intractable because it requires evaluating the joint probability over exponentially many behavior combinations. We show that Gibbs sampling provides an efficient solution. The algorithm repeatedly (i) selects a context, (ii) resamples its behavior conditioned on all other current behaviors, and (iii) updates the d-policy. Under mild conditions, this converges to a distribution concentrated on high-coherence d-policies (Theorem 4.2).

**Contributions.** We make three main contributions:

1. We show how debate, bootstrap, and ICM relate to coherence optimization. Debate is an exact instance, while simple bootstrap approximates coherence optimization near $\beta = 1$ (Proposition 4.5) and ICM optimizes a bidirectional variant of the objective (Equation 3).

2. We prove that among description-length regularizers for semi-supervised learning, coherence regularization with a pretrained prior optimizes a lower bound of worst-case accuracy. Preliminary experiments show that coherence-based metrics outperform LLM-as-a-judge as proxies for truthfulness.

3. We present a simple, scalable, and theoretically grounded algorithm for general coherence optimization.

## 2 Related Works

This section reviews literature on the formal and experimental study of feedback-free self-improvement.

**Unsupervised Scalable Oversight and Feedback-Free Self-Improvement.** Scalable oversight (Bowman et al., 2022), an area of AI alignment research (Ji et al., 2025), focuses on directly or indirectly supervising strong and potentially superhuman models, ones capable of gaming any available supervision signal. Classical methods of scalable oversight include debate (Irving et al., 2018; Brown-Cohen et al., 2024; Khan et al., 2024), iterated amplification (Wu et al., 2021), recursive reward modeling (Leike et al., 2018), and

weak-to-strong generalization (Kenton et al., 2024). Some of these methods, including debate, are unsupervised or self-supervised, where the aim is to uplift the capabilities of a model using its own supervision, without sacrificing safety properties. Newer methods in this category, including internal coherence maximization (Wen et al., 2025) and sampling methods based on Markov chain Monte Carlo (Karan & Du, 2025), have managed to match supervised finetuning performance on golden labels, a surprising feat for unsupervised methods. To date, the study of such methods has remained heuristic and entirely empirical — with rare exceptions studying the computational complexity of debate (Brown-Cohen et al., 2024; 2025) — and it is generally not understood why such methods work. This paper aims to change that by formally characterizing the mechanism of such feedback-free self-improvement (i.e., that these methods are policy-wide description-length regularization), and proves that coherence is the optimal regularization scheme for semi-supervised learning. It also gives a practical and easily scalable algorithm and demonstrates its empirical promise. The ICM algorithm (Wen et al., 2025) is closely related to the formalism of coherence optimization. It optimizes for the bidirectional variant of the autoregressively defined objective in coherence optimization, as will be discussed in §4.2. It is exactly this difference, however, that enables the scalable optimization and formal analysis of the autoregressive objective in this paper, which was difficult or intractable for ICM.

**Theory of Semi-Supervised Learning and Regularization.** The theory of semi-supervised learning characterizes how knowledge of the marginal distribution $P(x)$ of the input $x$ restricts the effective complexity of learning the conditional output distribution $P(y|x)$. Consistency regularization is a classical type of semi-supervised learning method, shown to reduce the local Rademacher complexity of the hypothesis class when applied on unlabeled data and lead to tighter generalization bounds (Maximov et al., 2018). While it is famously shown that unlabeled data improves sample complexity by at most a constant factor in the worst case (Ben-David et al., 2008), that constant factor depends on the data dimension and can be large in high-dimensional regimes. For instance, in sparse Gaussian mixtures, unlabeled data enables polynomial-time learning where supervised methods are computationally intractable (Azar & Nadler, 2024), and in the case of large neural networks, the neural scaling law varies with the intrinsic dimension of the data manifold (Sharma & Kaplan, 2022). However, high dimensionality can also be a curse; dimensional collapse — where embeddings degenerate into a low-rank subspace due to implicit regularization — is a common failure mode for semi-supervised learning (Jing et al., 2022), which He et al. (2024) propose countering via orthogonality regularization. In the deep learning regime, consistency regularization is shown to be equivalent to spectral decomposition of the augmentation graph (HaoChen et al., 2021). Wei et al. (2021) provided guarantees for self-training under an expansion assumption, a condition recently adapted to explain how strong student models can generalize beyond weak supervisors (Lang et al., 2024). At the information-theoretic limit, these regularizers can be modeled with Solomonoff induction (Leike, 2016), a connection formalized with singular learning theory to link loss landscape degeneracy to model compressibility (Urdshals et al., 2025). Our work contributes to this literature by (i) generalizing consistency regularization from invariance under local $\epsilon$-perturbations to predictability under arbitrary context differences; and (ii) suggesting that in the high-dimensional setting of language models, coherence regularization optimally reaps the large, dimensionality-dependent improvement factor from unlabeled data access, as upper-bounded by Ben-David et al. (2008).

**Formal Accounts of Reflective Equilibrium.** Interestingly, another line of related work comes from philosophical epistemology, which gives a conceptual and formal description of the possible *end states* of coherence optimization. Reflective equilibrium, first introduced in Rawls (1971), refers to the process of iteratively revising one's principles and case judgments to remove inconsistencies between them, until eventually converging upon the state of equilibrium. While the original proposal cleanly distinguishes principles from judgments, there have been generalized proposals considering beliefs lying on the full spectrum of generality (Rawls, 2005). Attempts to formalize reflective equilibrium with the language of AI started with Beisbart (2021), which built a formal model of reflective equilibrium with a mix of classical logic and optimization formalism, while stochastic elements were later introduced in Dellsén (2024). Prompted by theoretical interest in the question of when reflective equilibria of individuals converge in a group (Tersman, 2024), simulation studies have emerged with a diversity of setup design, including in the space of formal logic with logical inference distances as proximity measure (Lohse, 2023) and in the space of simple classifiers with classification margins as proximity measure (Baumgaertner & Lassiter, 2024). There also exists theoretical debate on, to paraphrase in machine learning language, whether the mutual support in a reflective equilibrium should be

autoregressive (Daniels, 1996; Holmgren, 1989) or bidirectional (DePaul, 2006; Haslett, 1987), with concerns that the latter leads to circularity; we revisit this question in §4, when comparing ICM and coherence optimization. All the works above have been limited to highly simplistic belief spaces and usually without any element of *learning*, which makes such formalisms inappropriate for actual application in machine learning. We aim to change this by introducing an expressive, learning-based formalism that comes directly with practical algorithms implemented on large language models, and provides grounding for reflective equilibrium by associating it, both theoretically and experimentally, with increased accuracy and truthfulness.

## 3 Defining Coherence

We develop a framework for understanding coherence optimization as a form of regularization in semi-supervised learning. We begin with the abstract problem, then introduce learning systems as a tractable way to implement it.

### 3.1 The Semi-Supervised Learning Problem

Consider a semi-supervised learning setup with behavior space $\mathcal{A}$ (a finite set of all possible behaviors) and context partition $\mathcal{S} \in \Pi(\mathcal{A})$ (a partition of $\mathcal{A}$ into sets of competing behaviors). Each context $s \in \mathcal{S}$ is a set of mutually exclusive behaviors, and each behavior $a \in \mathcal{A}$ belongs to exactly one context.

> **Definition 3.1** (Deterministic Policy)**.** A *deterministic policy* (d-policy) is a function $\pi : \mathcal{S} \to \mathcal{A}$ such that $\pi(s) \in s$ for all $s \in \mathcal{S}$. The space of all d-policies is $\mathcal{A}^{\mathcal{S}} := \prod_{s \in \mathcal{S}} s$.

Let $\mathcal{D} \in \Delta\left[\mathcal{A}^{\mathcal{S}}\right]$ be the data-generating distribution over d-policies. Let $\mathcal{F} \in \Delta[\mathcal{S}]$ be the context distribution, and $\{(s_n, a_n)\}_{n=1}^{N}$ the supervised samples (context-behavior pairs with ground-truth labels). We assume unlimited access to unsupervised samples (contexts in $\mathcal{F}$, without labels).

Our goal is to find a d-policy $\pi \in \mathcal{A}^{\mathcal{S}}$ that maximizes accuracy:

$$\alpha(\pi) := \mathrm{Pr}_{s \sim \mathcal{F}, \pi^* \sim \mathcal{D}} \left[\pi(s) = \pi^*(s)\right].$$

Empirical risk minimization (ERM) overfits when the hypothesis space is too large and expressive. The classical remedy is structural risk minimization (SRM) with description-length regularization. Given a prior distribution $\mathcal{P} \in \Delta\left[\mathcal{A}^{\mathcal{S}}\right]$ over d-policies,

$$\pi_{\mathrm{SRM}} := \underset{\pi \in \mathcal{A}^{\mathcal{S}}}{\arg\max} \ \alpha_{\mathrm{train}}(\pi) - \mathrm{reg}(\pi), \ \text{where} \tag{1}$$

$$\alpha_{\mathrm{train}}(\pi) := \frac{1}{N} \sum_{n=1}^{N} \mathbf{1}_{\pi(s_n)=a_n},$$

$$\mathrm{reg}(\pi) := \left(\frac{-2\log_2 \mathcal{P}(\pi) + \log_2 e + \log_2(1/\delta)}{2N}\right)^{1/2},$$

and $\mathcal{P}(\pi)$ is the probability mass of d-policy $\pi$ under prior $\mathcal{P}$.

We will show in §5 that the optimal choice of prior $\mathcal{P}$ for SRM is the one that minimizes $\mathrm{KL}[\mathcal{D}\|\mathcal{P}]$, i.e., the best available approximation to the true data-generating distribution. For language models, this is the pretrained model. §5.2 formalizes this intuition and proves that coherence regularization with a pretrained prior is optimal among all description-length regularizers in a worst-case asymptotic sense.

### 3.2 Learning Systems as Tractable Priors

The optimal prior $\mathcal{P}$ is the best approximation to the data-generating distribution, but we still need to compute probabilities under it. A prior over d-policies assigns probability to each of the $\prod_{s \in \mathcal{S}} |s|$ possible d-policies, and direct enumeration is intractable.

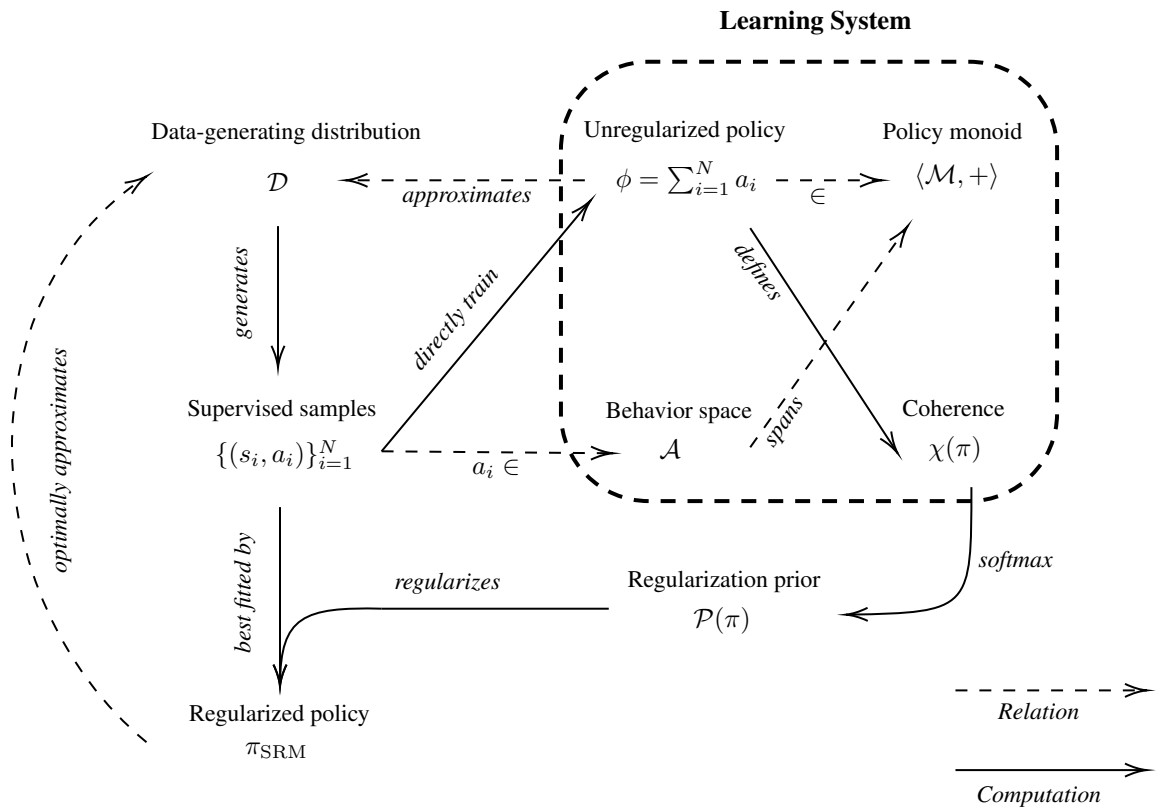

Figure 1: The coherence optimization framework. The data-generating distribution $\mathcal{D}$ produces supervised samples. Coherence optimization finds $\pi_{\mathrm{SRM}}$ maximizing empirical accuracy with regularization from prior $\mathcal{P}$. A learning system $(\mathcal{M}, \mathcal{A}, \mathcal{S}, \sigma)$ defines the coherence function $\chi$, providing a tractable instance of $\mathcal{P}$ via softmax over coherence: $\mathcal{P}(\pi) = \mathrm{X}^\beta(\pi) \propto 2^{\beta\chi(\pi)}$. Bayesian inference, in-context learning (ICL), and finetuning are instances of learning systems.

We need priors that allow efficient computation. A *learning system* $(\mathcal{M}, \mathcal{A}, \mathcal{S}, \sigma)$ defines such a prior, where $\mathcal{M}$ is a space of stochastic policies (e.g., training datasets or in-context example sets) and $\sigma$ is an inference function. The learning system specifies how to compute d-policy probabilities autoregressively, as the probability of behavior $a_1$ in the context $s_1$, times the probability of $a_2$ in $s_2$ conditioned on $(s_1, a_1)$, and so on. The log of this joint probability thereby equals what we call the *coherence* of the d-policy. By optimizing for coherence, we maximize $\log_2 \mathcal{P}(\pi)$, and thereby minimize $\mathrm{reg}(\pi)$ in Equation 1. For now, we will thus focus on characterizing and optimizing for coherence.

---

**Definition 3.2** (Learning System). A learning system is a quadruple $(\mathcal{M}, \mathcal{A}, \mathcal{S}, \sigma)$, where the *policy monoid* $\langle \mathcal{M}, + \rangle$ is a commutative monoid with the identity element $0$ (the *base policy*), a finite generating set $\mathcal{A} \subset \mathcal{M}$ of $\mathcal{M}$ (the *behavior space*), a partition $\mathcal{S} \in \Pi(\mathcal{A})$ of the behavior space $\mathcal{A}$ (the *context partition* or the *competing behavior partition*), and the *inference function* $\sigma$ mapping every $(\phi, s) \in \mathcal{M} \times \mathcal{S}$ to a distribution over $s$ that satisfies the *chain rule*:

$$\sigma(\phi, s_1)(a_1) \cdot \sigma(\phi + a_1, s_2)(a_2) = \sigma(\phi, s_2)(a_2) \cdot \sigma(\phi + a_2, s_1)(a_1), \quad \forall a_1 \in s_1 \in \mathcal{S}, a_2 \in s_2 \in \mathcal{S}.$$

---

The notation in Definition 3.2 directly extends §3.1. The behavior space $\mathcal{A}$ and context partition $\mathcal{S}$ are the same objects, and a d-policy $\pi \in \mathcal{A}^{\mathcal{S}}$ satisfies $\pi(s) \in s$ for all $s \in \mathcal{S}$. The main purpose of learning systems is to provide a tractable class of priors $\mathcal{P}$ for SRM, namely via the *softmax over coherence* (defined below).

Intuitively, every element in the behavior space is a "sample" with a certain input-output behavior, and every stochastic policy $\phi \in \mathcal{M}$ can be represented by a multiset of samples, i.e., its "training data". Each context $s \in \mathcal{S}$ is a set of competing behaviors. The chain rule asks that the cumulative "loss" of sequentially training on multiple samples is not affected by the ordering of samples.

---

**Example 3.3** (Bayesian Learning System)**.** A Bayesian learning system is defined by

$\langle \mathcal{M}, + \rangle$ Let a latent variable $\theta \in \Theta$ govern the data-generating process. A stochastic policy $\phi \in \mathcal{M}$ corresponds to a posterior distribution $\Pr[\theta \mid D_\phi]$ over $\Theta$, obtained by conditioning a base prior $\Pr[\theta]$ on a multiset of observations $D_\phi$. The base policy 0 corresponds to the prior $\Pr[\theta]$ (i.e., $D_0 = \emptyset$). The commutative monoid $\langle \mathcal{M}, + \rangle$ is the space of these observation multisets, with multiset union as the operation $+$.

$\mathcal{A}$ The set of all possible observations or data points.

$\mathcal{S}$ A partition of $\mathcal{A}$ where each context $s \in \mathcal{S}$ represents a set of mutually exclusive outcomes for a single experiment or query.

$\sigma$ The Bayesian predictive distribution. For a stochastic policy $\phi$ (representing data $D_\phi$) and a context $s$, the probability of observing a specific behavior $a \in s$ is its marginal likelihood: $\sigma(\phi, s)(a) = \Pr[a \mid D_\phi] = \int_{\theta \in \Theta} \Pr[a \mid \theta]\Pr[\theta \mid D_\phi]\mathrm{d}\theta$.

The chain rule holds as a direct consequence of the definition of joint probability. The LHS is

$$\sigma(\phi, s_1)(a_1) \cdot \sigma(\phi + a_1, s_2)(a_2) = \Pr[a_1 \mid D_\phi]\Pr[a_2 \mid D_\phi, a_1] = \Pr[a_1, a_2 \mid D_\phi],$$

while the RHS is

$$\sigma(\phi, s_2)(a_2) \cdot \sigma(\phi + a_2, s_1)(a_1) = \Pr[a_2 \mid D_\phi]\Pr[a_1 \mid D_\phi, a_2] = \Pr[a_2, a_1 \mid D_\phi].$$

Since joint probability is symmetric, the rule holds.

---

**Example 3.4** (In-Context Learning System)**.** An in-context learning system is defined by

$\langle \mathcal{M}, + \rangle$ The policy space $\mathcal{M}$ is the set of all possible multisets of in-context examples. The base policy 0 is the empty multiset. The operation $+$ is multiset union, which is commutative by definition. In practice, these multisets are linearized into sequences to be fed into a language model, and this linearization breaks strict commutativity; hence, this is an approximation.

$\mathcal{A}$ The set of individual input-output pairs that can be used as in-context examples, e.g., (context, behavior) pairs.

$\mathcal{S}$ Each context $s \in \mathcal{S}$ is the set of possible behaviors for a given input. For instance, for an input $q_s$, $s = \{(q_s, a) \mid a \in \text{possible behaviors}\}$.

$\sigma$ For a stochastic policy $\phi \in \mathcal{M}$ (a multiset of examples) and a context $s$ corresponding to input $q_s$, $\sigma(\phi, s)(a)$ for $a = (q_s, \tilde{a}) \in s$ is the probability assigned by the language model to generating behavior $\tilde{a}$ given the prompt constructed from $\phi$ and $q_s$. That is, $\sigma(\phi, s)(a) = P_{\text{LM}}(\tilde{a}|\text{context}(\phi), q_s)$.

Note that this example defines in-context learning as a *learning system*, which provides a tractable prior for regularization. It does not claim that in-context learning is itself coherence optimization. The claim in §4.2 is that methods like debate, bootstrap, and ICM optimize coherence as defined *through* such learning systems.

For in-context learning systems, it is unclear how well the commutativity of $\langle \mathcal{M}, + \rangle$ and the chain rule hold in practice. However, to the extent that in-context learning is an approximation of Bayesian inference (Xie

---

et al., 2021), such assumptions make sense as a first-order approximation. Also, in practice, we only face in-context sequences that are relatively well-mixed, so order dependency issues caused by a big in-context "distribution shift" over the turns is not a severe concern here.

---

**Example 3.5** (Finetuning Learning System). A finetuning learning system is defined by. . .

$\langle \mathcal{M}, + \rangle$ The policy space $\mathcal{M}$ can be identified with the set of all possible multisets of training data. The base policy 0 corresponds to the empty dataset and a randomly initialized model. The operation $+$ is multiset union. A stochastic policy $\phi \in \mathcal{M}$ represents the dataset used to train a model from its initial state to its current state.

$\mathcal{A}$ The set of all possible labeled data points $(x, y)$.

$\mathcal{S}$ Each context $s \in \mathcal{S}$ corresponds to a single input $x_s$ and is the set of all possible labeled data points for that input: $s = \{(x_s, y) \mid y \in \text{possible labels}\}$.

$\sigma$ For a stochastic policy $\phi \in \mathcal{M}$ (representing training data $D_\phi$) that has produced a model with parameters $\theta_\phi$, and a context $s$ corresponding to input $x_s$, $\sigma(\phi, s)(a)$ for $a = (x_s, y) \in s$ is the probability the model assigns to label $y$ for input $x_s$. That is, $\sigma(\phi, s)(a) = P(y|x_s; \theta_\phi)$.

As with Example 3.4, this example defines finetuning as a learning system (a source of tractable priors), not as a form of coherence optimization itself.

For finetuning learning systems, it is unclear how well the commutativity of $\langle \mathcal{M}, + \rangle$ and the chain rule hold in practice. For instance, the order of training data matters for stochastic gradient descent. However, to the extent that supervised learning is an approximation of Bayesian inference (Mingard et al., 2021), such assumptions make sense as a first-order approximation. Also, as before, we only face training example sequences that are relatively well-mixed, so order dependency issues caused by big distribution shifts are not a severe concern here.

---

*Remark* 3.6 (On the Chain Rule Assumption). A formal bound on the approximation error of the chain rule for in-context learning and finetuning systems remains an open question. However, the practical algorithms for coherence optimization do not depend on the chain rule holding exactly. The Gibbs sampling algorithm (Algorithm 1) optimizes conditional predictability across contexts, and the mutual predictability objective (Equation 3) uses bidirectional conditioning. Both remain well-defined and meaningful even when the chain rule is only approximately satisfied. In particular, they do not reduce to greedy decoding, because they optimize joint predictability across the full set of contexts rather than marginal likelihood of individual behaviors. The chain rule is needed for the equivalence between coherence and joint log-probability (Definition 3.2), but the algorithms optimize conditional predictability directly and converge to high-quality d-policies regardless.

D-policies $\pi \in \mathcal{A}^{\mathcal{S}}$ are to be distinguished from stochastic policies $\phi \in \mathcal{M}$. The latter is probabilistic, mapping every context to a distribution over its behaviors via the inference function $\sigma$.

We now define coherence, the central quantity of this paper. Coherence measures how well the responses in a d-policy "fit together" according to the prior policy. A high-coherence d-policy is one where each response is predictable given the others.

---

**Definition 3.7** (Coherence). Given any *prior policy* $\phi \in \mathcal{M}$, for any d-policy $\pi$, its *coherence* $\chi_\phi(\pi)$ relative to $\phi$ is defined by

$$\chi_\phi(\pi) = \sum_{n=1}^{|\mathcal{S}|} \log_2 \sigma \left( \phi + \sum_{m=1}^{n-1} \pi(s_m), s_n \right) (\pi(s_n)) \leq 0$$

where $s_1, \cdots, s_{|\mathcal{S}|}$ are the contexts in $\mathcal{S}$ in arbitrary order. The chain rule ensures that $\chi_\phi(\pi)$ is well-defined, as it implies that one can always swap neighbouring contexts in the sequence $\{s_i\}$ without changing $\chi_\phi(\pi)$, and thus the ordering of $\{s_i\}$ does not affect $\chi_\phi(\pi)$. In particular, we denote $\chi(\pi) := \chi_0(\pi)$.

Note that coherence operates on the probabilities assigned by the inference function $\sigma$, not on string equality. If the prior model has learned that two behaviors are semantically equivalent (e.g., $P[\text{yes} \mid \text{yeah}] \approx P[\text{yes} \mid \text{yes}]$), they will receive similar conditional probabilities and thus contribute similarly to coherence. Semantic equivalence is handled to the extent that the prior policy treats semantically equivalent outputs similarly.

From now on, we will denote with $s_1, \cdots, s_{|\mathcal{S}|}$ an arbitrary ordering of contexts in $\mathcal{S}$.

---

**Definition 3.8** (Softmax Over Coherence). For any *temperature reciprocal* $\beta > 0$, the *softmax over coherence* $\mathrm{X}^\beta : \mathcal{A}^\mathcal{S} \to \mathbb{R}_{\geq 0}$ is a probability distribution over d-policies such that the probability mass $\mathrm{X}^\beta(\pi) \propto 2^{\beta \chi(\pi)}$. In particular, when $\beta = +\infty$, $\mathrm{X}^\beta$ collapses into a uniform distribution over $\arg\max_\pi \chi(\pi)$, with 0 probability mass everywhere else.

---

The softmax over coherence at $\beta = 1$ provides an instance of the prior $\mathcal{P}$ from §3.1. Setting $\mathcal{P} = \mathrm{X}^1$ gives $\mathcal{P}(\pi) \propto 2^{\chi(\pi)}$. This is the tractable prior that learning systems provide for description-length regularization.

With coherence and its softmax in hand, we can state the optimization problem that the rest of the paper studies.

---

**Definition 3.9** (Coherence Optimization). Given a prior policy $\phi \in \mathcal{M}$, *coherence optimization* is the problem of finding a d-policy of maximal coherence,

$$\pi^\star \in \arg\max_{\pi \in \mathcal{A}^\mathcal{S}} \chi_\phi(\pi).$$

Equivalently, it is the problem of drawing samples from the softmax over coherence $\mathrm{X}^\beta$ in the low-temperature limit $\beta \to +\infty$, whose mass concentrates on the maximizers of $\chi_\phi$.

---

The two formulations agree because $\mathrm{X}^\beta$ places all of its mass on $\arg\max_\pi \chi_\phi(\pi)$ as $\beta \to +\infty$ (Definition 3.8). We will work mostly with the second, sampling-based formulation, for reasons explained in §4.

---

**Example 3.10** (Sauces). Consider the following learning system. The context partition is $\mathcal{S} = \{s^{\text{burger}}, s^{\text{fries}}\}$, where $s^{\text{burger}} = \{a^{\text{burger}}_{\text{mayo}}, a^{\text{burger}}_{\text{mustard}}, a^{\text{burger}}_{\text{other}}\}$ contains behaviors in the context "What sauce do you use for burgers?" and $s^{\text{fries}} = \{a^{\text{fries}}_{\text{mayo}}, a^{\text{fries}}_{\text{ketchup}}, a^{\text{fries}}_{\text{other}}\}$ contains competing behaviors in the context "What sauce do you use for dipping French fries?"

Consider the Bayesian learning system $\mathcal{M}$ with the following prior over $s^{\text{burger}} \times s^{\text{fries}}$:

| | $a^{\text{burger}}_{\text{mayo}}$ | $a^{\text{burger}}_{\text{mustard}}$ | $a^{\text{burger}}_{\text{other}}$ |
|---|---|---|---|
| $a^{\text{fries}}_{\text{mayo}}$ | 0.3 | $\epsilon$ | $\epsilon$ |
| $a^{\text{fries}}_{\text{ketchup}}$ | $\epsilon$ | $0.175 - \epsilon$ | $0.175 - \epsilon$ |
| $a^{\text{fries}}_{\text{other}}$ | $\epsilon$ | $0.175 - \epsilon$ | $0.175 - \epsilon$ |

Given the symmetry, let us only calculate $\mathrm{X}^\beta$ for the two d-policies $\pi_1 = (a^{\text{burger}}_{\text{mayo}}, a^{\text{fries}}_{\text{mayo}})$ and $\pi_2 = (a^{\text{burger}}_{\text{mustard}}, a^{\text{fries}}_{\text{ketchup}})$. For simplicity, assume $\epsilon = 0$.

In a Bayesian learning system, the coherence $\chi(\pi)$ simplifies to the $\log_2$ joint probability of the d-policy under the prior. Let $s_1 = s^{\text{burger}}$ and $s_2 = s^{\text{fries}}$. Then

$$\begin{aligned}
\chi(\pi) &= \log_2 \sigma(0, s_1)(\pi(s_1)) + \log_2 \sigma(\pi(s_1), s_2)(\pi(s_2)) \\
&= \log_2 P(\pi(s_1)) + \log_2 P(\pi(s_2) | \pi(s_1)) \\
&= \log_2 \left( P(\pi(s_1)) \cdot \frac{P(\pi(s_1), \pi(s_2))}{P(\pi(s_1))} \right) \\
&= \log_2 P(\pi(s_1), \pi(s_2)).
\end{aligned}$$

---

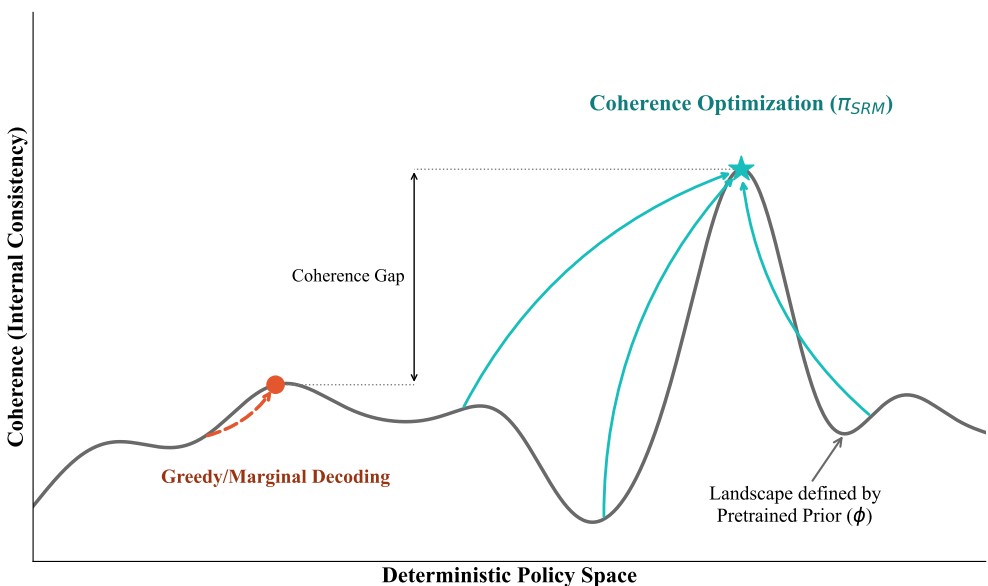

Figure 2: While greedy decoding gets trapped in local optima, coherence optimization finds the global peak of coherence/compressibility defined by the pretrained prior. Note the distinction between the coherence *landscape* (determined by the pretrained prior) from the *points* (deterministic policies in $\mathcal{A}^{\mathcal{S}}$).

---

Using the provided table with $\epsilon = 0$: for $\pi_1 = (a_{\text{mayo}}^{\text{burger}}, a_{\text{mayo}}^{\text{fries}})$, the joint probability is $P(a_{\text{mayo}}^{\text{burger}}, a_{\text{mayo}}^{\text{fries}}) = 0.3$. Thus, its coherence is $\chi(\pi_1) = \log_2(0.3) \approx -1.74$. For $\pi_2 = (a_{\text{mustard}}^{\text{burger}}, a_{\text{ketchup}}^{\text{fries}})$, the joint probability is $P(a_{\text{mustard}}^{\text{burger}}, a_{\text{ketchup}}^{\text{fries}}) = 0.175$. Thus, its coherence is $\chi(\pi_2) = \log_2(0.175) \approx -2.51$.

We see that $\chi(\pi_1) > \chi(\pi_2)$. This means the d-policy of using mayo for both burgers and fries is more coherent, even though mayo is not the most popular marginal choice for either food item. The marginal probabilities are $P(a_{\text{mayo}}^{\text{burger}}) = 0.3$ vs. $P(a_{\text{mustard}}^{\text{burger}}) = 0.35$, and $P(a_{\text{mayo}}^{\text{fries}}) = 0.3$ vs. $P(a_{\text{ketchup}}^{\text{fries}}) = 0.35$.

---

*Remark* 3.11 (What is coherence?). The sauces example might make coherence seem trivial at first. Why do we need another fancy name for joint probability? The value of such concept will be much clearer in an in-context learning system, and, even more so, in a finetuning learning system. In these setups, coherence is no longer reducible to well-known existing concepts and is hard to optimize for directly.

For an in-context learning system, coherence equals the cumulative perplexity when generating a list of d-policy responses *sequentially*. Note that greedy decoding at temperature 0 does *not* optimize such a cumulative perplexity (neither in theory nor in practice); the only decoding method that does it in theory is beam search with an infinite number of beams.

For a finetuning learning system, coherence equals the cumulative cross-entropy loss when training the prior policy on a list of d-policy responses *sequentially*. Despite the definition, however, it's almost certainly intractable to directly perform gradient descent on such a coherence function to optimize for it.

Given these challenges in optimizing for coherence, how should we proceed?

## 4 Algorithms for Coherence Optimization

Having defined coherence, we now turn to the question of its optimization. Figure 2 illustrates the goal of coherence optimization.

### 4.1 Gibbs Sampling for Coherence Optimization

Recall from Definition 3.9 that we want a d-policy of maximal coherence. Solving for $\arg\max_\pi \chi_\phi(\pi)$ directly is intractable, since the search ranges over the $\prod_{s\in\mathcal{S}} |s|$ d-policies and $\chi_\phi$ ties all contexts together, so the behavior chosen in one context cannot be optimized independently of the others. We therefore take a classical route and trade the maximization for a sampling problem. Rather than search for the mode of the softmax over coherence $\mathrm{X}^\beta \propto 2^{\beta\chi}$, we sample from it with Gibbs sampling (Geman & Geman, 1984) and raise the inverse temperature $\beta$ so that the samples concentrate on the high-coherence d-policies. This is the same device that underlies sampling-based optimization across statistics and machine learning, from the Metropolis algorithm (Metropolis et al., 1953; Hastings, 1970) and simulated annealing (Kirkpatrick et al., 1983) to learning in Boltzmann machines (Ackley et al., 1985), and the relationship between maximizing a probability and sampling from it is the familiar one between maximum a posteriori estimation and posterior sampling (Robert & Casella, 2004). We use this machinery in its standard form. The inverse temperature $\beta$ plays the role it plays in simulated annealing, interpolating between near-uniform sampling at small $\beta$ and maximization as $\beta \to +\infty$.

We now state the algorithm.[3]

---

**Algorithm 1** Gibbs Sampling Over D-Policies

---

1: **Input:** Learning system $(\mathcal{M}, \mathcal{A}, \mathcal{S}, \sigma)$; initial d-policy $\pi_0$; number of steps $N$; temperature reciprocal $\beta$.
2: **Output:** List of d-policies $\pi_0, \pi_1, \cdots, \pi_N$.
3: **procedure** GIBBSSAMPLE
4:     **for** $t = 0, 1, \ldots, N-1$ **do**
5:         Sample $n \sim \mathrm{Unif}[1..|\mathcal{S}|]$.
6:         Sample $a_t^n \sim \sigma^\beta \left( \sum_{m\neq n} \pi_t(s_m), s_n \right)$.
7:         Set $\pi_{t+1}$ such that $\pi_{t+1}(s_n) = a_t^n$ and $\pi_{t+1}(s_m) = \pi_t(s_m)$ for $m \neq n$.
8:     **end for**
9: **end procedure**

---

The algorithm is simple to implement (repeatedly resampling behaviors using behaviors in other contexts as condition) and easy to scale (see Algorithm 2 for a variant optimized for concurrency). A variant of it was implemented by Xu et al. (2023) for the different purpose of learning Chain-of-Thought reasoning.

We show below that Algorithm 2 approximately maximizes coherence.

> **Definition 4.1** (Ergodic Learning System). A learning system $(\mathcal{M}, \mathcal{A}, \mathcal{S}, \sigma)$ is *ergodic* if for every proper subset $C$ of the d-policy space $\mathcal{A}^{\mathcal{S}}$, there exist d-policies $\pi \in C$ and $\pi' \in \mathcal{A}^{\mathcal{S}} \setminus C$ such that $\pi'$ agrees with $\pi$ under all contexts except one $\dot{s} \in \mathcal{S}$, and $\sigma(\sum_{s\neq\dot{s}} \pi(s), \dot{s})(\pi'(\dot{s})) > 0$. In particular, all Bayesian learning systems with a full-support prior are ergodic (see Appendix B for proof).

> **Theorem 4.2** (Gibbs Sampling Recovers Softmax Over Coherence). *For ergodic learning system $(\mathcal{M}, \mathcal{A}, \mathcal{S}, \sigma)$, initial d-policy $\pi_0$, and temperature reciprocal $\beta > 0$, when $N \to +\infty$, we have, for Algorithm 1,*
> $$\pi_r \xrightarrow{d} \mathrm{X}^\beta, \quad \text{where } r \sim \mathrm{Unif}(\{0, \cdots, N\}).$$

*Proof.* See Appendix B. $\square$

Theorem 4.2 is the standard convergence guarantee for Gibbs sampling specialized to our setting, and its proof in Appendix B is the usual detailed-balance argument, restated for completeness. We claim no novelty for the sampler or for this convergence result. What is specific to this paper is the learning-system

---

[3]In Algorithm 1, the '$\beta$' in $\sigma^\beta$ means applying the transformation $p \to p^\beta$ to all probability masses and re-normalizing them.

construction that turns $\chi_\phi$ into a well-defined and computable objective, together with the reading of existing self-improvement methods as coherence optimization (§4.2).

*Remark* 4.3 (Tractability and mixing). Convergence to $X^\beta$ does not by itself mean that we can maximize coherence quickly. Theorem 4.2 is an asymptotic statement about the stationary distribution, and reaching that distribution in a useful number of rounds requires the chain to mix fast. The mixing time is controlled by the spectral gap of the transition kernel, which can shrink as $\beta$ grows and the landscape of $\chi_\phi$ becomes sharply peaked or multimodal, so very low temperatures can be slow to explore. When we call coherence optimization tractable, we mean that each Gibbs step is cheap, namely one conditional resample of a single context, and that the chain reaches high-coherence d-policies within a feasible number of rounds in our experiments. We do not mean that worst-case global maximization is guaranteed in polynomial time. Raising $\beta$ too aggressively is the annealing analogue of the mode collapse and over-optimization that we observe empirically in §6, where the chain locks onto a narrow region before it has explored enough. We mitigate this in practice by keeping $\beta$ moderate and by early stopping, as discussed in §6.

### 4.2 Reductions from Existing Methods of Self-Improvement

Having formulated a general and tractable algorithm for coherence optimization, we now show how state-of-the-art feedback-free self-improvement methods relate to it. These relationships range from an exact reduction in the case of debate, to an approximation in the case of simple bootstrap, to a closely related variant in the case of ICM. We make the nature of each connection precise below, rather than claiming that every method is a strict special case.

**Debate.** Debate (Irving et al., 2018) is an unsupervised method for scalable oversight where copies of a language model argue for opposing propositions in a back-and-forth structure. It has been shown to significantly increase language model accuracy on factual tasks (Li et al., 2022; Du et al., 2023), and allows for effective weak-to-strong truthfulness supervision (Khan et al., 2024).

The connection to coherence optimization is direct. Debate is Gibbs sampling with $|\mathcal{S}| = 2$. Consider a learning system with two contexts, $\mathcal{S} = \{s_{\mathrm{pro}}, s_{\mathrm{con}}\}$, representing arguments for and against a proposition $P$. The iterative debate process, where each side updates its argument based on the other's, is formally identical to Algorithm 1. Each debater resamples their argument conditioned on their opponent's current argument, exactly as Gibbs sampling resamples one component of the d-policy conditioned on all others.

> **Proposition 4.4** (Debate Maximizes Coherence). *Iterative debate for a learning system with $|\mathcal{S}| = 2$ is a synchronous variant of Gibbs Sampling (Algorithm 1). For an ergodic learning system, the distribution of d-policies $\pi_t$ converges to the softmax over coherence $X^\beta$.*

Appendix C.1 gives a formal specification and algorithm pseudocode of iterative debate. Proposition 4.4 then follows by tautology.

**Simple Bootstrap.** Bootstrap methods, where the output of a model is used for training itself, have been reported to increase accuracy without supervision as early as in Lee et al. (2013). It is recently revisited in the language modeling context as an unsupervised *inference-time* technique, where it matches the performance of supervised finetuning on golden labels (Chen et al., 2023).

Simple bootstrap can be seen as an approximation of Gibbs sampling where history accumulates rather than being replaced. Instead of maintaining a fixed-size d-policy and resampling components, simple bootstrap sequentially generates behaviors $a_1, a_2, a_3, \ldots$, where the context for generating $a_n$ is the entire preceding sequence.

> **Proposition 4.5** (Simple Bootstrap Approximates Coherence). *Let the sequence of contexts $(s_1, \ldots, s_N)$ be a permutation of $\mathcal{S}$ chosen uniformly at random. The total variation distance $D_{\mathrm{TV}}(P_{\mathrm{SB}}, X^\beta)$ between the simple bootstrap distribution and the softmax over coherence satisfies $D_{\mathrm{TV}} = O(|\beta - 1|)$ as $\beta \to 1$.*

*Proof.* See Appendix C.2. □

The $O(|\beta - 1|)$ approximation error limits the usefulness of this approximation at low temperatures ($\beta \gg 1$), where Gibbs sampling remains effective but simple bootstrap may not.

**Internal Coherence Maximization.** Internal coherence maximization (ICM) (Wen et al., 2025) optimizes for mutual predictability between model responses on different contexts through a hill-climbing process. Like simple bootstrap, it matches the performance of golden label finetuning.

Coherence and mutual predictability (the ICM objective) are closely related:

$$\chi(\pi) = \sum_{n=1}^{|\mathcal{S}|} \log_2 \sigma \left( \sum_{m<n} \pi(s_m), s_n \right) (\pi(s_n)), \tag{2}$$

$$f_{\mathrm{MP}}(\pi) = \sum_{n=1}^{|\mathcal{S}|} \log_2 \sigma \left( \sum_{m \neq n} \pi(s_m), s_n \right) (\pi(s_n)). \tag{3}$$

The difference is that coherence uses autoregressive conditioning ($m < n$) while mutual predictability uses bidirectional conditioning ($m \neq n$). Bidirectional conditioning is more expensive to compute, which is why ICM is restricted to domains with few discrete responses (e.g., multiple-choice questions), while Gibbs sampling over coherence scales to open-ended generation. See Appendix C.3 for further analysis.

## 5 Optimality of Coherence Optimization

In this section, we establish that coherence optimization is a form of description-length regularization, and that, in a worst-case asymptotic sense, it optimizes a lower bound of worst-case accuracy among description-length regularizers. We are careful below to state exactly what this optimality does and does not establish.

### 5.1 Coherence as Description-Length Regularization

The concept of coherence has a natural and powerful interpretation, specifically the principle of minimum description length (MDL). The MDL principle states that the best model for a set of data is the one that permits the shortest description of the data.

Consider a d-policy $\pi$ as a dataset, where each $\pi(s_n)$ is a data point. The coherence $\chi(\pi)$ is defined as a sum of sequential log-probabilities:

$$\chi(\pi) = \log_2 \sigma(0, s_1)(\pi(s_1)) + \log_2 \sigma(\pi(s_1), s_2)(\pi(s_2)) + \dots$$

This is precisely the log-probability of observing the sequence of behaviors $(\pi(s_1), \pi(s_2), \dots)$ under the predictive model defined by the learning system. By the chain rule, this is equal to the joint log-probability, $\log_2 \Pr[\pi(s_1), \dots, \pi(s_{|\mathcal{S}|})]$. By arithmetic coding, negated coherence is, when rounded up, the *number of bits needed to describe the d-policy.*

Coherence optimization, therefore, acts as a form of regularization that favors d-policies embodying strong, compressible patterns, and forces the model to discover and adhere to underlying principles rather than memorizing a set of unrelated behaviors. We can thus prove that coherence optimization, like other types of regularization, gives a uniform convergence guarantee on the d-policy's generalization error.

> **Definition 5.1** (Agreement and Accuracy). For any two d-policies $\pi_1, \pi_2 \in \mathcal{A}^{\mathcal{S}}$ and a subset of contexts $\hat{\mathcal{S}} \subseteq \mathcal{S}$, we define the *agreement* between $\pi_1$ and $\pi_2$ on $\hat{\mathcal{S}}$ as
>
> $$\alpha_{\hat{\mathcal{S}}}(\pi_1; \pi_2) = \alpha_{\hat{\mathcal{S}}}(\pi_2; \pi_1) := \frac{1}{|\hat{\mathcal{S}}|} \sum_{s \in \hat{\mathcal{S}}} \mathbf{1}_{\pi_1(s) = \pi_2(s)}.$$

In particular, we denote $\alpha(\pi_1; \pi_2) := \alpha_{\mathcal{S}}(\pi_1; \pi_2)$. Given any *ground-truth d-policy* $\pi^*$, we define d-policy $\pi$'s *accuracy* on $\hat{\mathcal{S}}$ as $\alpha_{\hat{\mathcal{S}}}(\pi; \pi^*)$.

**Theorem 5.2** (Uniform Convergence). *Let $\pi^*$ be any given ground-truth d-policy, and the* training contexts *$\hat{s}_1, \hat{s}_2, \cdots, \hat{s}_N$ be uniformly and independently sampled contexts from $\mathcal{S}$, for some $N > 0$. Let $\pi$ be an arbitrary d-policy with $\alpha_{\text{train}}(\pi; \pi^*) := \alpha_{\{\hat{s}_i\}_{i=1}^N}(\pi; \pi^*)$, then*

$$|\alpha(\pi; \pi^*) - \alpha_{\text{train}}(\pi; \pi^*)| \leq \sqrt{\frac{-2\chi(\pi) + \log_2 e + \log_2(1/\delta)}{2N}}$$

*holds uniformly for all $\pi$ with probability $1 - \delta$, for any given $\delta \in (0, 1)$.*

Theorem 5.2 tells us that, the coherence of a d-policy helps bound its generalization gap, i.e., the difference between its training and actual accuracy. But recall that coherence can be defined with respect to any prior policy; does the choice of prior policy affect the generalization gap? The answer is yes.

**Definition 5.3** (Optimality Gap). Given any ground-truth d-policy $\pi^*$, for any stochastic policy $\phi \in \mathcal{M}$, we define the *optimality gap* of $\phi$ as the prior policy to be

$$G(\phi; \pi^*) := -2\chi_\phi(\pi^*) + \log_2 e > 0.$$

**Proposition 5.4** (Optimality Gap Lower-Bounds Accuracy). *Under the conditions of the uniform convergence theorem, for any stochastic policy $\phi \in \mathcal{M}$, let*

$$\hat{\pi} := \arg\max_{\pi \in \mathcal{A}^{\mathcal{S}}} \left\{ \alpha_{\text{train}}(\pi; \pi^*) - \sqrt{\frac{-2\chi_\phi(\pi) + \log_2 e + \log_2(1/\delta)}{2N}} \right\}, \tag{4}$$

*then, with probability $1 - \delta$,*

$$\alpha(\hat{\pi}; \pi^*) \geq 1 - \sqrt{\frac{2G(\phi; \pi^*) + 2\log_2(1/\delta)}{N}}. \tag{5}$$

Proposition 5.4 tells us that the goodness of a prior policy is decided by the coherence of the optimal d-policy under such a prior, where goodness is defined as the actual accuracy of the optimal d-policy when using such a prior for coherence regularization.

Note that both Definition 5.3 and Proposition 5.4 assume there is one fixed $\pi^*$, rather than a $\pi^*$ drawn from a distribution $\mathcal{D} \in \Delta[\mathcal{A}^{\mathcal{S}}]$, as is the case in §3.1. However, Proposition 5.4 can be easily adapted to the latter case, where instead of Equation 5, we have

$$\mathrm{E}_{\pi^* \sim \mathcal{D}}\left[\alpha(\hat{\pi}; \pi^*)\right] \geq 1 - \mathrm{E}_{\pi^* \sim \mathcal{D}}\left[\sqrt{\frac{2G(\phi; \pi^*) + 2\log_2(1/\delta)}{N}}\right].$$

## 5.2 Worst-Case Asymptotic Optimality of Coherence Regularization

The uniform convergence theorem tells us that coherence bounds the generalization gap. But which prior should we use? We now prove that the KL-optimal approximation to the data-generating distribution is the optimal choice.

Consider the semi-supervised learning setup from §3, with context partition $\mathcal{S}$, behavior space $\mathcal{A}$, d-policy space $\mathcal{A}^{\mathcal{S}}$, data-generating distribution $\mathcal{D} \in \Delta\left[\mathcal{A}^{\mathcal{S}}\right]$, context distribution $\mathcal{F} \in \Delta[\mathcal{S}]$, and $N$ supervised

samples. Under SRM with description-length regularization using prior $\mathcal{P}$, with

$$\pi_{\text{SRM}} := \arg\max_{\pi \in \mathcal{A}^{\mathcal{S}}} \ \alpha_{\text{train}}(\pi) - \text{reg}(\pi), \ \text{where}$$

$$\text{reg}(\pi) := \left( \frac{-2\log_2 \mathcal{P}(\pi) + \log_2 e + \log_2(1/\delta)}{2N} \right)^{1/2},$$

we have the following optimality result.

---

**Theorem 5.5** (Optimality of Description-Length Regularization). *Under description-length regularization with prior $\mathcal{P}$, for any distribution $\mathcal{Q} \in \Delta[\mathcal{A}^{\mathcal{S}}]$ and sufficiently small $\delta$ such that $\log_2(1/\delta) \gg -\mathrm{E}_{\pi \sim \mathcal{Q}}[\log_2 \mathcal{P}(\pi)]$ (i.e., we consider worst-case outcomes at the lowest quantiles), with probability $1 - \delta$,*

$$\alpha(\pi_{\text{SRM}}) \geq \mathrm{E}_{\pi \sim \mathcal{Q}}[\alpha(\pi)] - \sqrt{\frac{2\log_2(1/\delta)}{N}} + \sqrt{\frac{2 + o(1)}{N\log_2(1/\delta)}} \ (\mathrm{H}\,[\mathcal{Q}] - \mathrm{KL}\,[\mathcal{Q}\,||\,\mathcal{P}]). \tag{6}$$

*In particular, when $\mathcal{Q} = \mathcal{D}^{\beta}$ for $\beta > 0$,[a] the lower bound is maximized when $\mathcal{P}$ minimizes $\mathrm{KL}\left[\mathcal{D}^{\beta}\,||\,\mathcal{P}\right]$.*

---
[a]$\mathcal{D}^{\beta}$ is the distribution obtained by raising $\mathcal{D}$'s probability masses to the $\beta$-th power before normalizing.

---

*Proof.* See Appendix D.4. □

Practically, let $\hat{\mathcal{D}}$ be the KL-optimal approximation to $\mathcal{D}$ learnable from the supervised samples, then, for $\mathcal{Q} = \mathcal{D}^{\beta}$, the optimal obtainable prior is $\mathcal{P} = \hat{\mathcal{D}}^{\beta}$, in line with Definition 3.8 when using the supervised policy as prior policy. This is why pretrained models are the natural choice of prior for coherence regularization. A pretrained language model, trained on the largest and most representative sample of $\mathcal{D}$, is the best available KL-approximation and therefore yields the tightest worst-case accuracy guarantee from Theorem 5.5.

The bound requires $\log_2(1/\delta) \gg -\log_2 \mathcal{P}(\pi)$, meaning we consider worst-case outcomes at the lowest quantiles (small $\delta$). This is a strong assumption that may not hold when $\delta$ is not sufficiently small relative to the d-policy's description length. The next section examines the hypothesis that when the prior policy is trained on supervised samples, optimizing coherence alone is equivalent to optimizing the regularized training objective above.

**What this optimality does and does not show.** Theorem 5.5 should be read carefully. It says that coherence regularization with the KL-optimal prior maximizes a particular lower bound on accuracy, in the regime where $\delta$ is small and we are looking at the lowest quantiles of outcomes. Maximizing a lower bound is weaker than being the best regularizer in practice. One can take any valid lower bound, subtract a nonnegative term from it, and call the result optimal for the modified bound, so optimality of a lower bound is only as meaningful as the bound is tight. Even when the bound is tight, it is tight in the worst case, and a method can be worst-case optimal while still being beaten on typical instances, which are rarely the worst case. We therefore read the result as a worst-case guarantee that motivates the use of a pretrained prior, not as a proof that coherence regularization dominates every alternative regularizer. A direct empirical comparison against other regularizers under matched compute would be needed to make the stronger claim, and we have not run one. We return to this gap in the Limitations.

### 5.3 Coherence with Trained Prior as Regularized Accuracy

The previous section established that coherence optimization is equivalent to description-length regularization. Proposition 5.4 tells us to optimize $\alpha_{\text{train}}(\pi) - \text{reg}(\pi)$, where $\text{reg}(\pi)$ depends on coherence. But in practice (e.g., in Algorithm 1), we often optimize coherence alone, without an explicit training accuracy term. Why does this work?

The answer is that the prior policy already encodes the training samples. When we use a pretrained or supervised-trained model as the prior, optimizing coherence with respect to this prior implicitly optimizes

a joint objective that includes training accuracy. We hypothesize that the two formulations, (i) optimizing $\alpha_{\text{train}}$ + coherence regularization and (ii) optimizing coherence with a trained prior, are asymptotically equivalent under appropriate conditions.

We want to be clear about the status of this section before stating it formally. The worst-case optimality result of the previous section (Theorem 5.5) is proven, and the optimality claim in the abstract and introduction rests on that theorem alone. The equivalence stated below is a separate claim about the average case, namely about when optimizing coherence with a trained prior, with no explicit training-accuracy term, recovers the regularized training objective. We state it as a conjecture and support it with a heuristic argument rather than a proof. Nothing elsewhere in the paper depends on it. Its purpose is to explain why optimizing coherence alone works in practice, and to suggest how to choose the number of unsupervised contexts.

To formalize this, we need machinery for reasoning about policies that have undergone two stages of training, namely pretraining and post-training.

---

**Lemma 5.6** (Change-of-Prior). *Denote*

$$\hat{\chi}_\phi[a_1, a_2, \cdots, a_k] := \sum_{n=1}^{k} \log_2 \sigma \left( \phi + \sum_{m=1}^{n-1} a_m, s_n \right) (a_n),$$

*where $a_n \in s_n$ for all $n$. For any policy $\rho \in \mathcal{M}, \phi = \rho + \sum_{n=1}^{N} a_n^\phi \in \mathcal{M}$ ($a_n^\phi \in \mathcal{A}$) and $\psi = \phi + \sum_{l=1}^{L} a_l^\psi \in \mathcal{M}$ ($a_l^\psi \in \mathcal{A}$), we have*

$$\hat{\chi}_\phi[a_1^\psi, \cdots, a_L^\psi] + \hat{\chi}_\rho[a_1^\phi, \cdots, a_N^\phi] = \hat{\chi}_\rho[a_1^\phi, \cdots, a_N^\phi, a_1^\psi, \cdots, a_L^\psi].$$

---

*Proof.* See Appendix D.4. □

---

**Definition 5.7** (Quotient System). For a learning system $(\langle \mathcal{M}, + \rangle, \mathcal{A}, \mathcal{S}, \sigma)$, its *quotient system spanned by $\mathcal{S}_a \subseteq \mathcal{S}$ at $\phi \in \mathcal{M}$* is a learning system $(\langle \mathcal{M}_a, +_a \rangle, \mathcal{A}_a, \mathcal{S}_a, \sigma_a)$ defined by three properties. First, $\mathcal{A}_a = \{\phi + u : u \in \bigcup_{s \in \mathcal{S}_a} s\}$. Second, $\mathcal{M}_a = \langle \mathcal{A}_a \rangle_{+_a}$ with identity element $0_a := \phi$, and the operator $+_a$ satisfying $(\phi + u) +_a (\phi + v) = \phi + (u + v)$. Third, $\sigma_a(\phi + u, s) = \sigma(\phi + u, s)$.

We denote such a relation with $(\langle \mathcal{M}_a, +_a \rangle, \mathcal{A}_a, \mathcal{S}_a, \sigma_a) \overset{\phi}{\trianglelefteq} (\langle \mathcal{M}, + \rangle, \mathcal{A}, \mathcal{S}, \sigma)$. We will denote with $\chi^a$ the coherence function in the quotient system.

---

Essentially, the quotient system spanned by $\mathcal{S}_a$ at $\phi$ represents the learning problem of post-training a pretrained policy $\phi$ on the context set $\mathcal{S}_a$. In the rest of this section, we will use "pretrain samples" to refer to our supervised samples, and "posttrain samples" to refer to our unsupervised samples, in line with the semi-supervised learning setup.

---

**Theorem 5.8** (Prior Policy Encodes Previous Training Samples). *Let*

$$(\mathcal{M}_a, \mathcal{A}_a, \mathcal{S}_a, \sigma_a) \overset{0_a}{\trianglelefteq} (\mathcal{M}, \mathcal{A}, \mathcal{S}, \sigma)$$

$$(\mathcal{M}_b, \mathcal{A}_b, \mathcal{S}_b, \sigma_b) \overset{0_b}{\trianglelefteq} (\mathcal{M}, \mathcal{A}, \mathcal{S}, \sigma)$$

*such that $\mathcal{S}_a \cap \mathcal{S}_b = \emptyset$, $\mathcal{S}_a \cup \mathcal{S}_b = \mathcal{S}$, and, for some $\pi \in \mathcal{A}^{\mathcal{S}}$, $0_a = \sum_{s \in \mathcal{S}_b} \pi(s)$ and $0_b = \sum_{s \in \mathcal{S}_a} \pi(s)$. Then:*

$$\underbrace{\chi^a(\pi(\mathcal{S}_a))}_{\substack{\text{posttrain coherence} \\ \text{w.r.t. pretrain prior}}} + \underbrace{\hat{\chi}_0[\pi(\mathcal{S}_b)]}_{\text{pretrain coherence}} = \chi(\pi) = \underbrace{\hat{\chi}_0[\pi(\mathcal{S}_a)]}_{\text{posttrain coherence}} + \underbrace{\chi^b(\pi(\mathcal{S}_b))}_{\substack{\text{posttrained accuracy} \\ \text{on pretrain samples}}}. \quad (7)$$

---

*Proof.* Both equalities follow from direct applications of the change-of-prior lemma. $\qquad\square$

One way to interpret this theorem is that it shows the equivalence between:

**LHS** Optimizing solely for a policy's coherence on posttrain (unsupervised) contexts wrt pretrained (supervised) prior,[4] *and*

**RHS** Optimizing jointly for the policy's coherence on posttrain (unsupervised) contexts and its accuracy on pretrain (supervised) samples.

Note that for the interpretation of RHS, there are still subtle gaps between the intuitive understanding here and the terms in equation 7, which we will revisit in Appendix D.5, in the heuristic argument for the conjecture below. We now conjecture the conditions under which optimizing coherence alone is equivalent to optimizing the regularized training objective.

> **Conjecture 5.9** (Equivalence of Coherence Optimization and Regularized Training)**.** *Let the training contexts $\pi$ be randomly separated into two disjoint subsets, posttrain samples $\pi(\mathcal{S}_a)$ and pretrain samples $\pi(\mathcal{S}_b)$. The latter is given, while we need to optimize the former (which can be any d-policy in $\prod_{s \in \mathcal{S}_a} s$). I.e., we do* not *know the ground-truth labels for the posttrain contexts, and our task is to determine them.*
>
> *Consider (i) optimizing for coherence on the posttrain samples $\pi(\mathcal{S}_a)$ with respect to a pretrained policy or (ii) optimizing for $\alpha_{\text{train}}(\pi) - \text{reg}(\pi)$, i.e., SRM. The two are asymptotically equivalent when the posttrain sample count $|\mathcal{S}_a|$ is chosen to satisfy*
>
> $$\underbrace{|\mathcal{S}_a|}_{\text{posttrain sample count}} \sim \frac{1}{4} \times \underbrace{\frac{\overbrace{\left(-\chi^b(\pi^*(\mathcal{S}_b))/|\mathcal{S}_b|\right)^2}^{\text{mean pretrain coherence, squared}}}{-\hat{\chi}_0[\pi^*(\mathcal{S}_a)]/|\mathcal{S}_a|}}_{\text{mean posttrain coherence, inversed}} \times \underbrace{\left(\frac{1}{1-\alpha^b(\pi^*)}\right)^2}_{\substack{\text{pretrain error rate,}\\ \text{inverse-squared}}} \times \underbrace{|\mathcal{S}_b|}_{\text{pretrain sample count}} \quad (8)$$

The argument we give for the conjecture in Appendix D.5 is heuristic. It relies on two approximations that we do not prove in general. The first is that cumulative training loss is roughly linear in accuracy times sample size, which we expect to hold for naive-Bayes-like models and approximately for language-model finetuning past the initial phase, but which we do not establish here. The second is that the relevant per-context quantities concentrate as the number of contexts grows. Turning the conjecture into a theorem would require making both approximations precise, for instance by fixing a generative model in which the loss-to-accuracy relationship is exact and the concentration can be controlled. We leave this to future work and use the conjecture only for intuition and for the hyperparameter guidance below.

The conjecture states that two formulations are equivalent when the number of posttrain samples $|\mathcal{S}_a|$ is chosen appropriately. Even if we do not trust the exact estimate in Equation 8, we can use ternary search to find the optimal $|\mathcal{S}_a|$, since concavity of $f(r)$ and $g(r)$ implies that $\max_{r \in \mathbb{R}}(f(r) - \sqrt{|\mathcal{S}_a|} \cdot g(r))$ is unimodal in $|\mathcal{S}_a|$. In this sense, $|\mathcal{S}_a|$ is a hyperparameter whose optimal value we need to search for.

The equivalence explains why optimizing coherence alone works in practice. When we use a pretrained language model as the prior policy, we are implicitly optimizing for the same samples that prior policy was trained on. The prior encodes the training objective; optimizing coherence with respect to this prior recovers the regularized training objective.

The conjecture also suggests that the optimal number of posttrain samples scales with the effective number of pretrain samples that are i.i.d. with the posttrain contexts. This is likely much larger than typical in-context learning implementations (Xu et al., 2023; Wen et al., 2025), which already produce strong results. This motivates supervised training-based implementations of coherence optimization, which allow greatly expanding the number of unsupervised samples; we discuss the corresponding method in Appendix B.2.

---

[4]The "pretrain coherence" term can be ignored as it's fixed.

**Coherence optimization as transductive learning.** Practical applications of coherence optimization are typically in-sample by design, similar to transductive learning. The goal is to find the best d-policy over a given set of contexts, and the "generalization" is from the supervised subset to the unsupervised subset of that same context set. Theorem 5.2 bounds accuracy on the full context set, which includes the unsupervised portion. In this sense, the unsupervised contexts serve simultaneously as the regularization medium and the evaluation target. This makes train/test splits less relevant than in standard supervised learning, because coherence optimization already evaluates each context relative to all others. Nevertheless, we also verify in Appendix E that coherence optimization improves accuracy on fully held-out test questions that were never part of the Gibbs sampling context.

**Practical implications.** The results in this section yield three actionable guidelines. First, the prior should be as close to the data-generating distribution as possible (Theorem 5.5). In practice, this means using a pretrained or supervised-trained model, and preferring priors that have been trained on data similar to the target task. Second, the number of unsupervised contexts for coherence optimization should be comparable to the effective number of i.i.d. supervised samples (Conjecture 5.9). Too many unsupervised contexts can overwhelm the supervised signal; too few limit the regularization benefit. Third, coherence overoptimization is mitigable through early stopping, temperature tuning, or limiting the number of unsupervised contexts (Appendix 6.4).

## 6 Preliminary Experiments

We present preliminary experiments demonstrating that coherence is effective both as an evaluation metric and as a training objective. We also present practical results comparing types of prior policies, which agree with Theorem 5.5 on the data-vs-prior distributional distance being a determinant of effectiveness. Current results are preliminary and should be treated as a starting point for exploration.

### 6.1 Coherence as an Evaluation Metric

We test the scalability of coherence as an unsupervised proxy for truthfulness by using a 0.6B weak judge model/prior policy to scrutinize GPT-5 outputs twice, with classical LLM-as-a-Judge and with a coherence-based metric respectively.

**Setup.** Each data point represents a collection of around 5 HARDMath questions (Fan et al., 2024) with strong topical relevance. Blue points represent GPT-5-generated honest solutions to a collection, while red points represent GPT-5-generated deceptive solutions (where GPT-5 is explicitly prompted to generate "incorrect but convincing" solutions). In Figure 3(a) (LLM-as-a-Judge), the y-axis represents the sum of logprobs the Qwen3-0.6B weak judge model assigns to the solutions. In Figure 3(b) (coherence), the y-axis represents the difference between cross-solution coherence and sum of individual logprobs, i.e., pointwise mutual information (PMI), using the same weak judge as prior policy. The x-axis in both plots represents solution length.

**Results.** Note that this evaluation is inherently cross-question. Each solution's coherence is computed relative to other solutions in the same topical collection, so the metric already evaluates each output in the context of others rather than in isolation.

The coherence-based metric turns out much superior, as shown in Figure 3. If PMI (the difference between coherence and sum of logprobs) is positively predictive of honesty, it implies coherence provides additional predictive power over LLM-as-a-Judge. Indeed, deceptive solutions receive *higher* logprobs in LLM-as-a-Judge, meaning the judge is fooled by deception. They receive *lower* PMIs, meaning the coherence-based metric detects deception attempts better than LLM-as-a-Judge does. This demonstrates that cross-context internal coherence predicts truthfulness, even when the content is incomprehensible to the judge.

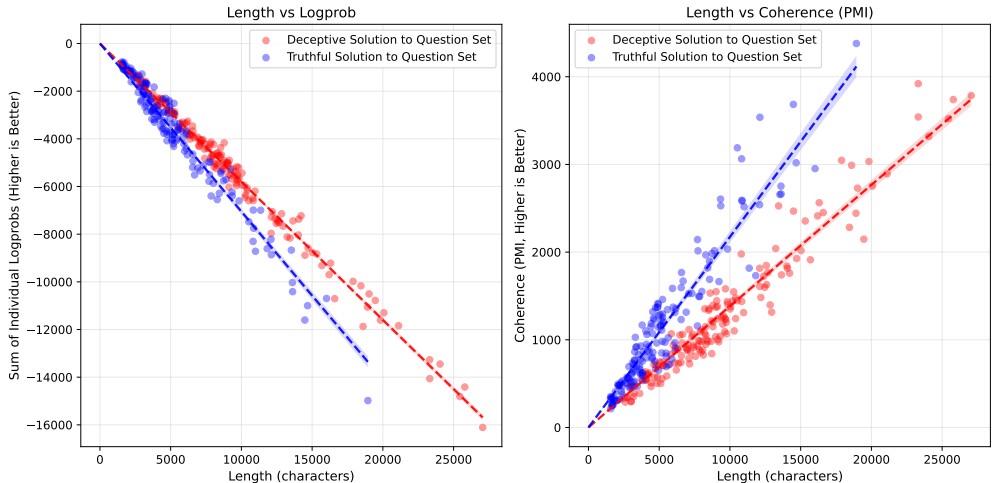

Figure 3: Coherence-based metrics detect deceptive solutions while LLM-as-a-Judge is fooled. Blue: honest solutions; Red: deceptive solutions. Left: LLM-as-a-Judge is fooled by deception and assigns higher probabilities (sum of logprobs) to deceptive solutions. Right: The coherence-based metric (PMI) successfully assigns higher scores to honest solutions.

## 6.2 Coherence as an Optimization Objective

We test whether coherence optimization improves GSM8K performance for a 1B-parameter LLM by implementing Algorithm 2, across $\gamma \in \{0.25, 0.45, 0.65, 0.85\}$. We evaluate each setup over 100 random seeds and report the mean results.

**Setup.** We use `meta-llama/Llama-3.2-1B-Instruct` in an in-context learning (ICL) learning system and run $N = 40$ Gibbs rounds on a per-run training set $\mathcal{S}$ of size $|\mathcal{S}| = 100$. Accuracy is exact-match on the final numeric answer (parsed from the model output). We evaluate each configuration over 100 random seeds; for each seed we draw a fresh 100-example subset $\mathcal{S}$ from GSM8K and run the full $\gamma$ sweep on this same subset, enabling paired comparisons across $\gamma$. Furthermore, we introduce a slightly modified sampling rule: for each $t \geq 1$ and each $s_n \notin \hat{\mathcal{S}}_t$, instead of sampling from $\sigma^\beta(\phi_t, s_n)$ we sample from an equal-weight mixture of the samplers induced by the base and current round contexts $(\phi_0, \phi_t)$:

$$a_t^n \sim \tfrac{1}{2}\,\sigma^\beta(\phi_t, s_n)\ +\ \tfrac{1}{2}\,\sigma^\beta(\phi_0, s_n).$$

In preliminary runs, sampling exclusively from the current-round context $\phi_t$ led to unstable dynamics: we observed occasional mode collapse and substantially higher variance across seeds. We theorize Gibbs updates can over-optimize toward a narrow region of high-coherence/low-performance $\pi$ distributions, amplifying early-round noise and reducing exploration.

To mitigate this, we anchor sampling to the initial context $\phi_0$ by drawing from an equal-weight mixture of $\sigma^\beta(\phi_t, \cdot)$ and $\sigma^\beta(\phi_0, \cdot)$. This can be viewed as a lightweight, training-free analogue of KL regularization. Empirically, this modification helped reduce mode collapse and yield more consistent trajectories while retaining performance across rounds.

**Results.** Figure 4 shows that coherence optimization consistently increases train accuracy over Gibbs rounds, with the strongest gains at smaller hold-out fractions. Across 100 paired seeds, $\gamma = 0.25$ achieves the best performance: mean train accuracy improves from 24.86% at initialization to 35.58% at the final round (+10.72%), with a peak of 45.90% on average (Table 1). As $\gamma$ increases, both the final and peak accuracies monotonically decline (peak: 41.56% at $\gamma = 0.45$, 38.40% at $\gamma = 0.65$, and 34.62% at $\gamma = 0.85$), suggesting that resampling a larger complement (smaller $\gamma$) provides a stronger self-training signal

Table 1: Aggregate summary statistics across Gibbs rounds. Mean $\pm$ SE across seeds.

| $\gamma$ | Rounds | Init Train (%) | Final Train (%) | Max Train (%) | Max Round | Improvement (%) |
|---|---|---|---|---|---|---|
| 0.25 | 40 | $24.86 \pm 0.48$ | $\mathbf{35.58 \pm 0.80}$ | $\mathbf{45.90 \pm 0.62}$ | $19.6 \pm 1.6$ | $\mathbf{+10.72 \pm 0.77}$ |
| 0.45 | 40 | $24.86 \pm 0.48$ | $33.56 \pm 0.60$ | $41.56 \pm 0.51$ | $19.1 \pm 1.5$ | $+8.70 \pm 0.68$ |
| 0.65 | 40 | $24.86 \pm 0.48$ | $31.86 \pm 0.64$ | $38.40 \pm 0.45$ | $20.7 \pm 1.5$ | $+7.00 \pm 0.66$ |
| 0.85 | 40 | $24.86 \pm 0.48$ | $27.72 \pm 0.64$ | $34.62 \pm 0.55$ | $20.4 \pm 1.6$ | $+2.86 \pm 0.72$ |

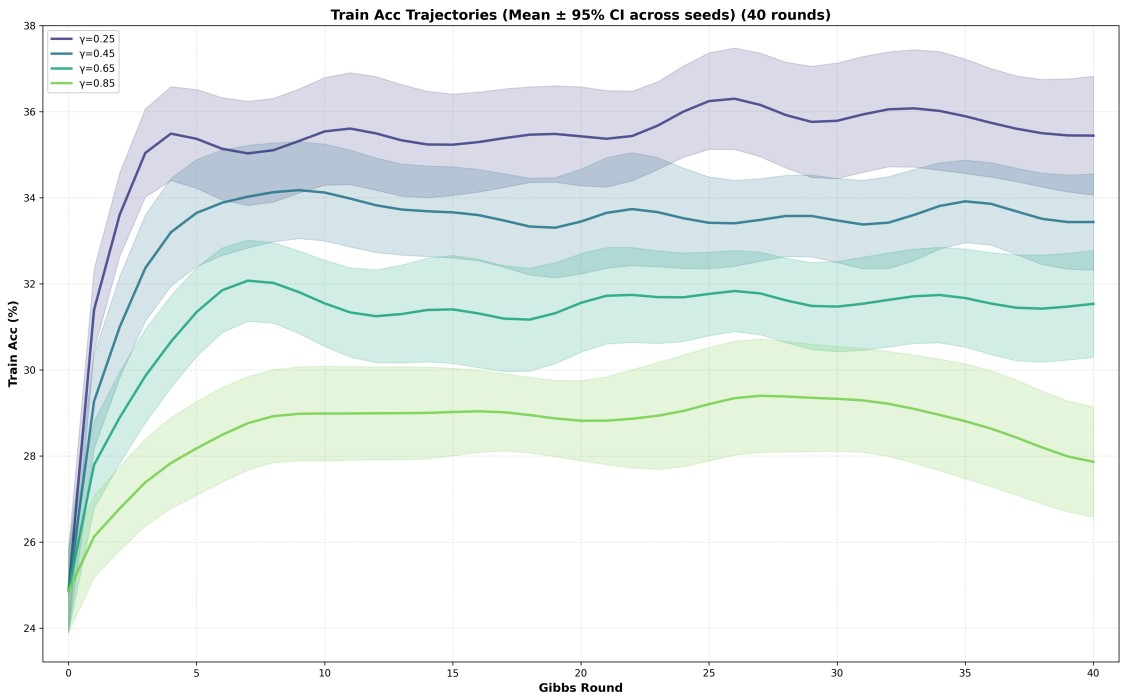

Figure 4: Mean train accuracy over 40 Gibbs rounds for $\gamma \in \{0.25, 0.45, 0.65, 0.85\}$ (shaded: 95% CI). Higher $\gamma$ holds out a larger context subset $\hat{\mathcal{S}}_t$ when resampling.

in this learning system. Notably, the round of maximal performance is relatively stable across $\gamma$ (roughly rounds 19-21 on average; Table 1), indicating that most gains occur in the first $\sim$20 rounds, after which improvement saturates. We hypothesize that this saturation arises from context-window limitations and inherent constraints of in-context learning (ICL), since the procedure can only improve predictions via contextual conditioning rather than explicit parameter updates.

While these results are in-sample, the observed saturation is consistent with a capacity ceiling: the `meta-llama/Llama-3.2-1B-Instruct` model reports 44.4% on GSM8K (Meta Llama, 2024). This is comparable in scale to our best average **in-sample** peak, but is not directly comparable due to differences in prompting/evaluation methods. **We report out-of-sample (test) accuracy on a separate set of** 200 **held-out GSM8K questions in Appendix E.** Test accuracy improves from roughly 30% to 40% under Gibbs sampling, confirming that coherence optimization generalizes beyond the in-sample context.

## 6.3 Effect of Prior Policy on Coherence Optimization

The choice of prior policy affects the performance of coherence optimization. We empirically examine how different types of priors influence the alignment between coherence and truthfulness.

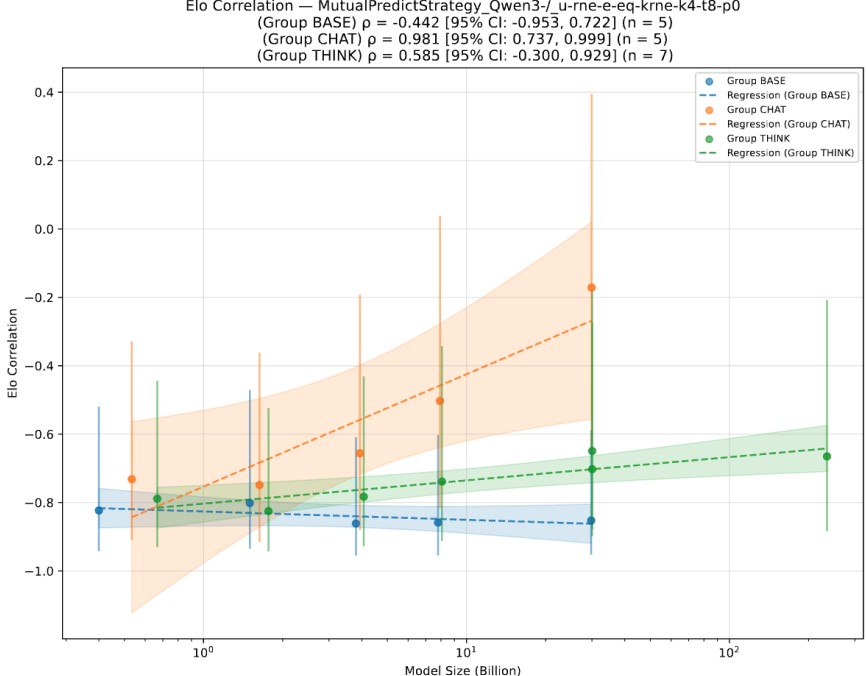

Figure 5: Scaling trends of coherence-truthfulness agreement for different prior policies in an in-context learning system. Lower is better. BASE refers to a pretrained prior, CHAT to a posttrained/RLHF prior, and THINK to a reinforced reasoning prior.

**Setup.** We compare three types of prior policies. (1) **Pretrained prior** (BASE) is a base model trained only on next-token prediction. (2) **Posttrained prior** (CHAT) is a model further trained with RLHF for helpfulness and safety. (3) **Reinforced reasoning prior** (THINK) is a model trained with RL on verifiable reasoning tasks. We use the Qwen3 model family for all three classes of prior policies.

Under each prior policy, and for each of 8 models chosen from LMArena (Chiang et al., 2024), we compute the negated coherence of generations from the latter model on a self-curated set of open-ended research questions. Then, for each prior policy, we compute the Pearson's r between a model's LMArena Elo rating and its negated coherence under that prior policy (shown on y-axis, lower is better), and examines how it changes w.r.t. the prior policy's parameter count (shown on x-axis). This would indicate the extent to which scaling is possible for coherence optimization, and the comparison of practical fit across types of prior policy.

**Results.** Figure 5 shows accuracy scaling trends for each prior type. Several patterns emerge.

When the model is small, all three prior types perform similarly, encoding basic grammar, logic, and commonsense. As model size increases, we observe different scaling slopes. A negative slope indicates that higher-fidelity modeling of that prior leads to better coherence-truthfulness alignment. We observe $slope_{BASE} < slope_{THINK} < slope_{CHAT}$. This suggests that adding the preference prior (CHAT) de-aligns coherence from accuracy, while further adding the factuality prior (THINK) re-aligns them.

The pretrained and reinforced reasoning priors follow the strategy of "using all available supervision to train a best policy, then let coherence optimization regularize with that policy as prior." This is favored by Theorem 5.5 in the sense that these priors best approximate the true data-generating distribution among the options available, and so give the tightest worst-case lower bound on accuracy.

### 6.4 Discussion

These experiments, while preliminary, reveal several patterns worth examining.

**Coherence Detects Deception When Direct Evaluation Fails.** The HARDMath results show that a weak judge's coherence-based metric outperforms its direct evaluation, even though both use the same model. How can comparing multiple outputs reveal information that evaluating a single output cannot?

A deceptive solution must simultaneously appear plausible to the judge and contain the deceptive content. The first condition is easy to satisfy in isolation, but the second introduces constraints that conflict with the model's learned distribution over related content. A deceptive mathematical argument must avoid certain true statements that would expose the deception. This avoidance makes the deceptive solution locally plausible but globally improbable when considered alongside topically related solutions. PMI captures this. High marginal probability but low joint probability indicates a solution optimized to appear good in isolation while avoiding broader context that would expose it.

An adversary can optimize for marginal plausibility, but optimizing for joint plausibility across many related contexts is equivalent to actually being truthful.

Note, however, the HARDMath experiment used synthetic deceptive solutions generated by prompting; it remains unclear whether the same signal would detect naturally occurring or adversarially trained deception.

**Coherence Overoptimization (Goodharting).** In practice, we sometimes see response texts becoming unnatural from the human perspective (yet still information-rich and score highly on automated benchmarks), after many rounds of Gibbs sampling. This may be due to limitations of in-context learning, where the objective is not exactly to imitate in-context examples, or it may be due to *overoptimization* of coherence. Below we briefly summarize the types of coherence overoptimization that are theoretically possible.

1. **Small divergence between desired measurement target (e.g. human legibility) and actual coherence measurement (e.g. pretrained policy).**

   Its impact on accuracy is modest and can be reflected by the increase in the optimality gap $G(\phi; \pi^*) = -2\chi_\phi(\pi^*) + \log_2 e$, whose continuity ensures that a small divergence only leads to small change in the gap size.

   However, its impact on alignment can be arbitrarily large, assuming that alignment is carried out by instilling a certain bias (e.g. human legibility) in the prior policy. As we have seen in Gao et al. (2023), optimizing for a flawed proxy for alignment bias can lead to catastrophic deviation from the desired bias (e.g. complete loss of human legibility).

2. **Too many coherence optimization contexts.** The informal theorem on joint optimization has determined the optimal number of contexts for coherence optimization to be on the same order as the effective number of i.i.d. supervised training samples. While such a number is hard to exceed in practice, when the coherence optimization contexts focus on a certain niche in the context space, they can easily outnumber the effective supervised samples in the same niche.

   Its impact on accuracy can be arbitrarily large. For instance, when the coherence optimization contexts are entirely orthogonal to supervised training samples, or only represent a vanishingly small portion of those samples, the optimality gap $G(\phi; \pi^*) = -2\chi_\phi(\pi^*) + \log_2 e$ goes to infinity.

   It has no negative impact on alignment, since alignment is itself embodied by the coherence optimization process, rather than being traded off against.

In both cases, to mitigate overoptimization, we may use early-stopping on three different aspects.

1. **Early stopping on Gibbs iterations**, i.e., to stop Gibbs sampling after a certain number of rounds when we see that desirable properties (e.g. accuracy, human legibility) start to decline.

2. **Reducing number of coherence optimization contexts**, typically to the same ballpark as the effective number of supervised learning samples.

3. **Higher temperature in the "softmax over coherence" distribution**, which in the Gibbs sampling case, implies a reduced inverse temperature $\beta$.

For all three interventions, we can use ternary search or similar methods to find the sweet spot, assuming the ability to evaluate desired properties (e.g. accuracy, human legibility).

# 7    Conclusion

This paper presented coherence optimization as a framework for feedback-free self-improvement. We showed how state-of-the-art methods relate to coherence optimization, with debate an exact instance and bootstrap and ICM close relatives, proved that coherence regularization with a pretrained prior optimizes a lower bound of worst-case accuracy among description-length regularizers, developed a principled method for coherence optimization, and demonstrated preliminary empirical evidence for its effectiveness.

**Appendix A addresses a list of frequently asked questions.**

### Limitations and Future Work

One major limitation concerns experimental validation. Our optimality result (Theorem 5.5) is theoretical, and it concerns a worst-case lower bound rather than typical-case performance, as discussed in §5.2. The experiments in §6 test elements of the theory such as coherence-truthfulness alignment, prior choice, and accuracy uplift, and the prior-comparison experiment is consistent with the theorem, but none of them validate the optimality claim by comparing coherence regularization head-to-head against other regularizers. We want to be explicit that the optimality claim is not yet empirically established. A fair comparison would match compute budgets across coherence regularization, entropy regularization, KL-divergence to a reference policy, and consistency regularization on a common semi-supervised task, and would report which one generalizes best rather than which one optimizes the bound we analyze. We see this as the most important direction for future work, and we are careful throughout the paper to phrase the optimality claim as a worst-case lower-bound result rather than a demonstrated empirical advantage. The number of benchmarks is also small, and broader evaluation is needed before drawing strong conclusions.

Additionally, our analysis suggests larger potential for coherence optimization if scaled to near-pretrain scale. By allowing a large reasoning model to perform extended reasoning on *all* the contexts in its pretraining data, it can discover the belief system that most coherently explains the world. This kind of coherence-seeking across otherwise unrelated contexts has given rise to many of humanity's most important realizations, e.g., special relativity arose from the inconsistency between the fixed speed of light and the behavior of fast-moving objects, and the immorality of slavery was recognized through the inconsistency between the institution and principles of innate human rights. We have also shown that two-agent debate is a special case of coherence optimization, and such debates occur in almost every instance of group deliberation. Coherence optimization thus serves as a principled model of reflection in both humans and AIs, and future work may explore its extension to multi-agent and human-AI systems.

### Broader Impact Statement

This work proposes coherence optimization as a framework for feedback-free self-improvement in language models. While the primary motivation is to improve model truthfulness without requiring external supervision, we acknowledge potential dual-use concerns. On the positive side, coherence optimization could enhance the reliability of AI systems by identifying and resolving internal inconsistencies, and develops a characterization of what self-improvement methods truly optimize for, both important for AI safety. On the negative side, any method that improves model capabilities could potentially be misused or contribute to loss-of-control risks. We believe the benefits of developing mechanistic understanding of unsupervised self-improvement outweigh risks brought by the capablity uplifts already mostly exhausted by existing methods.

### Acknowledgments

Tianyi Qiu and Zhonghao He are CI research fellows affiliated with the Oxford HAI Lab. We thank Callum Lawson, Dylan Hadfield-Menell, Fabien Roger, Jiaxin Wen, Aidan Ewart, Callum Canavan, and Max Kleiman-Weiner for helpful discussions.

Almost all substantial content is manually drafted, with editorial assistance from AI. Standard results, examples, straightforward explanation of setups, the illustrative Figure 2, and connective content are AI-drafted with manually-written outline, and undergoes extensive manual review and editing.

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

## A Frequently Asked Questions

In this section, we address potential questions about the overall framework.

### Does coherence optimization just find any "coherent-but-false" peak, e.g., conspiracy theories?

*Critique: Coherence optimization will find any peak of high internal consistency, even one that is factually wrong. It can't distinguish a "true" equilibrium from a "coherent-but-false" one.*

No. Coherence optimization does not happen in a vacuum. It is a *regularization technique* that functions on top of supervised signals, as established in the theoretical results (e.g., Proposition 5.4). Although the coherence optimization process itself uses no additional labels, the supervised signal enters through the pretrained prior, which encodes the knowledge from its training data. This is consistent with the semi-supervised framing in the introduction and §3.1.

- In Language Models: Coherence optimization inherits the vast knowledge embedded in the pretrained model. This prior *already* constrains the search space, making most "coherent-but-false" peaks (which contradict pretraining data) have a much lower coherence score from the start. Also, this distinguishability scales with reasoning strength; as we train stronger and stronger reasoner models, their ability to reason at length enhances immunity against hard-to-find incoherence that would not be obvious from just looking at the pretraining data.

- In Reinforcement Learning Agents: The agent receives Bernoulli feedback from the environment, which mixes both subjective coherence and the "supervised signal" to ground the coherence search.

It remains an open question whether these signals are sufficient to *uniquely* determine the globally optimal reflective equilibrium. However, they immensely narrow the search space. Historically, ideologies that are wrong in hindsight (e.g., the moral defense of slavery) are almost always revealed through their *incoherence against new, incoming observations* (e.g., the philosophical idea of universal human rights). Coherence optimization is the formal process for identifying and resolving exactly this type of inconsistency.

### Can we customize coherence to optimize for properties other than truthfulness?

*Critique: The paper focuses on truthfulness, but what if we care about other properties like human-readability or value alignment?*

Yes. The prior policy $\phi$ can be chosen to regularize for different traits. Recall that coherence $\chi_\phi(\pi)$ measures the log-probability of d-policy $\pi$ under prior $\phi$. By choosing $\phi$ appropriately, we can steer the optimization toward different goals:

- **Truthfulness:** Use a pretrained model as $\phi$. The pretrained model has seen vast amounts of factual data, so d-policies that contradict well-established facts will have low coherence.

- **Human legibility:** Use a model that predicts what humans find easy to understand as $\phi$. Since negated coherence equals description length, maximizing coherence under a "human prior" means minimizing the description length of the policy from a human perspective. This incentivizes behaviors that are predictable and interpretable to humans.

- **Value alignment:** Use an aligned model (e.g., one trained with RLHF on safety-focused data) as $\phi$. D-policies exhibiting misaligned behaviors, such as deception or sandbagging, will incur high description length under an aligned prior, since such behaviors are hard for an aligned model to predict.

In principle, any trainable model can serve as a prior, including human preference models. The role of coherence optimization is to turn such a prior (which evaluates individual outputs) into a policy-level regularizer (which evaluates the joint distribution of outputs across contexts).

**Why regularize at the policy level instead of per-output?**

*Critique: Standard methods like RLHF already regularize model outputs. What's the advantage of policy-level regularization?*

RLHF and similar methods apply supervision to the *marginal distribution*, rewarding or penalizing individual outputs in isolation. Policy-level regularization, by contrast, considers the *joint distribution* of outputs across all contexts.

This distinction matters because certain failure modes are invisible in marginals but obvious in joints. Figure 6 illustrates the problem. Given only marginal distributions (the "shadow" of each peak), you cannot tell which peak is tallest. But the joint distribution (the full 3D landscape) makes the answer obvious. Consider sandbagging, where a model performs poorly on some tasks but well on others. Each individual output might look reasonable in isolation; the problem only becomes apparent when comparing outputs across contexts. Deception works similarly. A lie in one context might seem plausible until you notice it contradicts what the model said elsewhere.

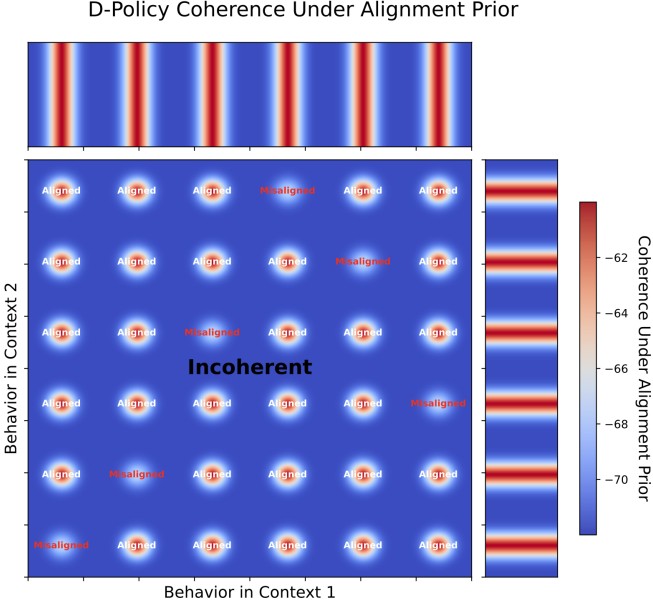

Figure 6: Illustration of d-policy space. Peaks represent coherent d-policies. Given only marginal distributions (the "shadow" of each peak), it is impossible to distinguish the tall peak from the short ones. Coherence optimization selects for high joint probability, naturally steering toward the tallest peak.

Coherence optimization naturally catches these failures. A d-policy that sandbags or deceives will have low coherence, because such behaviors are hard to predict from the model's other outputs. The inconsistency manifests as high description length.

**Does coherence optimization require a fully truthful model to begin with?**

*Critique: If we need a truthful prior to improve truthfulness, don't we already need to solve the problem first?*

Not necessarily. What we need is that the prior policy be *substantially truthful* in a precise sense. The optimality gap $G(\phi; \pi^*) = -2\chi_\phi(\pi^*) + \log_2 e$ of the ground-truth d-policy $\pi^*$ must be smaller than the expected optimality gap $\mathrm{E}_{\pi'}[G(\phi; \pi')]$ for a random d-policy $\pi'$ drawn from the prior. In other words, truth must outperform the average response under the prior, though it need not be the most likely answer to every individual question. The prior is allowed to sometimes produce falsehoods, and coherence optimization corrects such errors by favoring the globally most coherent d-policy. When this condition is violated, the generalization bound (Proposition 5.4) becomes vacuous and coherence optimization may converge to a

coherent-but-false equilibrium. This is exactly the failure mode the theory predicts. As practical strategies for ensuring this condition holds, we may consider the following.

1. **Interleaved training:** Start from a base pretrained model. Use it as the prior while simultaneously doing coherence optimization and capability training. The pretrained prior constrains the training to stay consistent with the factual knowledge already encoded.

2. **Weak-to-strong:** Use a smaller, well-understood model as the prior for training a larger model. The smaller model's factual grounding transfers to the larger model through coherence regularization, even though the smaller model is less capable.

The circularity is broken because coherence optimization amplifies this existing truthful tendency in pre-trained models rather than creating it from scratch.

# B  Examples and Proofs for Gibbs Sampling-Based Coherence Optimization

This section provides detailed examples and formal proofs for the Gibbs sampling-based coherence optimization methods.

## B.1  The Original Gibbs Sampling Method

---

**Example B.1** (Gibbs Sampling Over Sauces). Let us trace one full update step of the algorithm, starting from a plausible but non-optimal d-policy $\pi_0 = (a_{\text{mustard}}^{\text{burger}}, a_{\text{ketchup}}^{\text{fries}})$. Let the temperature reciprocal $\beta = 1$ (i.e., no temperature scaling).

**Sample the new burger behavior, $a_0^1$.** The context for this sample is the *other* component of the current d-policy, which is $\pi_0(s^{\text{fries}}) = a_{\text{ketchup}}^{\text{fries}}$. We sample from the distribution $\sigma(a_{\text{ketchup}}^{\text{fries}}, s^{\text{burger}})$, which in this Bayesian system is the conditional probability distribution $P(s^{\text{burger}}|a_{\text{ketchup}}^{\text{fries}})$. Using the prior table (with $\epsilon = 0$): First, find the marginal probability of the context: $P(a_{\text{ketchup}}^{\text{fries}}) = P(a_{\text{mayo}}^{\text{burger}}, a_{\text{ketchup}}^{\text{fries}}) + P(a_{\text{mustard}}^{\text{burger}}, a_{\text{ketchup}}^{\text{fries}}) + P(a_{\text{other}}^{\text{burger}}, a_{\text{ketchup}}^{\text{fries}}) = 0 + 0.175 + 0.175 = 0.35$. Next, compute the conditional probabilities for each burger choice: $P(a_{\text{mayo}}^{\text{burger}}|a_{\text{ketchup}}^{\text{fries}}) = P(a_{\text{mayo}}^{\text{burger}}, a_{\text{ketchup}}^{\text{fries}})/P(a_{\text{ketchup}}^{\text{fries}}) = 0/0.35 = 0$; $P(a_{\text{mustard}}^{\text{burger}}|a_{\text{ketchup}}^{\text{fries}}) = P(a_{\text{mustard}}^{\text{burger}}, a_{\text{ketchup}}^{\text{fries}})/P(a_{\text{ketchup}}^{\text{fries}}) = 0.175/0.35 = 0.5$; and $P(a_{\text{other}}^{\text{burger}}|a_{\text{ketchup}}^{\text{fries}}) = P(a_{\text{other}}^{\text{burger}}, a_{\text{ketchup}}^{\text{fries}})/P(a_{\text{ketchup}}^{\text{fries}}) = 0.175/0.35 = 0.5$. We then sample $a_0^1$ from this distribution: {mayo: 0, mustard: 0.5, other: 0.5}. Suppose we sample $a_0^1 = a_{\text{mustard}}^{\text{burger}}$.

**Sample the new fries behavior, $a_0^2$.** The context for this sample is $\pi_0(s^{\text{burger}}) = a_{\text{mustard}}^{\text{burger}}$. We sample from the distribution $\sigma(a_{\text{mustard}}^{\text{burger}}, s^{\text{fries}})$, which is $P(s^{\text{fries}}|a_{\text{mustard}}^{\text{burger}})$. The marginal probability of the context is $P(a_{\text{mustard}}^{\text{burger}}) = 0 + 0.175 + 0.175 = 0.35$. The conditional probabilities for each fries choice are: $P(a_{\text{mayo}}^{\text{fries}}|a_{\text{mustard}}^{\text{burger}}) = 0/0.35 = 0$; $P(a_{\text{ketchup}}^{\text{fries}}|a_{\text{mustard}}^{\text{burger}}) = 0.175/0.35 = 0.5$; and $P(a_{\text{other}}^{\text{fries}}|a_{\text{mustard}}^{\text{burger}}) = 0.175/0.35 = 0.5$. We then sample $a_0^2$ from {mayo: 0, ketchup: 0.5, other: 0.5}. We sample $a_0^2 = a_{\text{other}}^{\text{fries}}$.

**Construct the new d-policy.** The new d-policy is $\pi_1 = (a_0^1, a_0^2) = (a_{\text{mustard}}^{\text{burger}}, a_{\text{other}}^{\text{fries}})$. The system has transitioned to a new d-policy, which has the same coherence as the initial one: $\chi(\pi_1) = \log_2(0.175) \approx -2.51$.

The run has not converged yet, but we stop here for illustration. Notice that the most coherent d-policy $\pi_1^* = (a_{\text{mayo}}^{\text{burger}}, a_{\text{mayo}}^{\text{fries}})$ is an absorbing state. If we ever sample $a_{\text{mayo}}^{\text{burger}}$, the next sample for fries will be $a_{\text{mayo}}^{\text{fries}}$ with probability 1, as $P(a_{\text{mayo}}^{\text{fries}}|a_{\text{mayo}}^{\text{burger}}) = 1$. Symmetrically, if we sample $a_{\text{mayo}}^{\text{fries}}$, the next burger choice must be $a_{\text{mayo}}^{\text{burger}}$. The sampler cannot escape this state, which has the highest coherence. For an ergodic system (e.g., if $\epsilon > 0$), the sampler would not get stuck but would instead visit this high-coherence state more frequently than others, eventually sampling from the softmax distribution over all d-policies.

---

We see that Gibbs sampling tends to converge upon high-coherence d-policies. We now prove that Bayesian learning systems with nonzero priors satisfy the ergodicity condition (Definition 4.1).

**Example B.2** (Bayesian Learning Systems With Nonzero Priors Are Ergodic). Recall that a system is ergodic if for any two d-policies $\pi, \pi'$ that differ in only one context $\dot{s}$, the transition probability $\sigma(\sum_{s \neq \dot{s}} \pi(s), \dot{s})(\pi'(\dot{s}))$ is greater than zero.

In a Bayesian learning system, this transition probability is the conditional probability $P(\pi'(\dot{s})|\{\pi(s)\}_{s \neq \dot{s}})$. By the definition of conditional probability,

$$P(\pi'(\dot{s})|\{\pi(s)\}_{s \neq \dot{s}}) = \frac{P(\pi'(\dot{s}), \{\pi(s)\}_{s \neq \dot{s}})}{P(\{\pi(s)\}_{s \neq \dot{s}})}.$$

The d-policy $(\pi'(\dot{s}), \{\pi(s)\}_{s \neq \dot{s}})$ is simply $\pi'$. A Bayesian system with a nonzero prior assumes that the joint probability $P(\pi_{joint})$ is strictly positive for *any* full d-policy $\pi_{joint}$.

Therefore, the numerator $P(\pi')$ is greater than zero. The denominator $P(\{\pi(s)\}_{s \neq \dot{s}})$ is a marginal probability, calculated by summing the joint probabilities over all possible behaviors in $\dot{s}$:

$$P(\{\pi(s)\}_{s \neq \dot{s}}) = \sum_{a \in \dot{s}} P(a, \{\pi(s)\}_{s \neq \dot{s}}).$$

Since every term in this sum is strictly positive by assumption, the denominator is also strictly positive. Thus, the conditional probability is the ratio of two positive numbers, which is positive. This holds for any pair of d-policies differing by a single component, so the system is ergodic.

---

**Theorem B.3** (Gibbs Sampling Recovers Softmax Over Coherence). *For any ergodic learning system* $(\mathcal{M}, \mathcal{A}, \mathcal{S}, \sigma)$, *initial d-policy* $\pi_0$, *and temperature reciprocal* $\beta > 0$, *when* $N \to +\infty$, *we have*

$$\pi_r \xrightarrow{d} X^{\beta}, \quad \text{where } r \sim \text{Unif}(\{0, \cdots, N\})$$

---

*Proof.* The Gibbs sampling algorithm defines a time-homogeneous Markov chain on the finite state space of d-policies $\mathcal{A}^{\mathcal{S}}$. To show that the distribution of states converges to $X^{\beta}$, we need to show that the chain is ergodic and that $X^{\beta}$ is its unique stationary distribution.

**Ergodicity:** The chain is irreducible because the learning system is assumed to be ergodic. By definition, this means for any $\pi, \pi'$ differing by one component, the transition probability is positive. Since any d-policy can be reached from any other d-policy by a sequence of single-component changes, there is a non-zero probability path between any two states in the chain. The chain is also aperiodic; for example, there is a non-zero probability of resampling the same behavior for a component, allowing the chain to stay in the same state. An irreducible and aperiodic Markov chain on a finite state space is ergodic.

**Stationary Distribution:** We show that $X^{\beta}$ satisfies the detailed balance condition, which is a sufficient condition for being a stationary distribution. For any two states $\pi, \pi' \in \mathcal{A}^{\mathcal{S}}$, the condition is $X^{\beta}(\pi)P(\pi \to \pi') = X^{\beta}(\pi')P(\pi' \to \pi)$. It suffices to check this for states $\pi, \pi'$ that can be reached from one another in a single step, i.e., those that differ in at most one component. If $\pi = \pi'$, the condition is trivial. Let $\pi$ and $\pi'$ differ only in a single context, $\dot{s}$. Let $\pi(\dot{s}) = a$ and $\pi'(\dot{s}) = a'$. Let $\phi = \sum_{s \neq \dot{s}} \pi(s) = \sum_{s \neq \dot{s}} \pi'(s)$. The transition probability $P(\pi \to \pi')$ involves updating the component $\dot{s}$ to $a'$. The probability for this specific update is proportional to $\sigma^{\beta}(\phi, \dot{s})(a')$. We write the full transition probability, accounting for the multi-step nature of the update: $P(\pi \to \pi') = \frac{1}{|\mathcal{S}|} \frac{\sigma(\phi, \dot{s})(a')^{\beta}}{Z_{\phi}}$, where $Z_{\phi} = \sum_{\tilde{a} \in \dot{s}} \sigma(\phi, \dot{s})(\tilde{a})^{\beta}$ is the normalization constant.

The key insight is that the coherence $\chi(\pi)$ is independent of the ordering of contexts. We can therefore choose an ordering where $\dot{s}$ is the last element, $s_{|\mathcal{S}|}$. With this ordering, the coherence of $\pi$ is:

$$\chi(\pi) = \left( \sum_{n=1}^{|\mathcal{S}|-1} \log_2 \sigma \left( \sum_{m=1}^{n-1} \pi(s_m), s_n \right) (\pi(s_n)) \right) + \log_2 \sigma(\phi, \dot{s})(a)$$

And for $\pi'$:

$$\chi(\pi') = \left( \sum_{n=1}^{|\mathcal{S}|-1} \log_2 \sigma \left( \sum_{m=1}^{n-1} \pi(s_m), s_n \right) (\pi(s_n)) \right) + \log_2 \sigma(\phi, \dot{s})(a')$$

The terms in the parentheses are identical for $\pi$ and $\pi'$. Therefore,

$$\chi(\pi) - \chi(\pi') = \log_2 \sigma(\phi, \dot{s})(a) - \log_2 \sigma(\phi, \dot{s})(a')$$

Taking the exponential and raising to the power of $\beta$:

$$\frac{2^{\beta\chi(\pi)}}{2^{\beta\chi(\pi')}} = \frac{\sigma(\phi, \dot{s})(a)^\beta}{\sigma(\phi, \dot{s})(a')^\beta}$$

Now, let's check the detailed balance condition using $\mathrm{X}^\beta(\pi) \propto 2^{\beta\chi(\pi)}$:

$$\mathrm{X}^\beta(\pi)P(\pi \to \pi') \propto 2^{\beta\chi(\pi)} \cdot \sigma^\beta(\phi, \dot{s})(a') = 2^{\beta\chi(\pi)} \cdot \frac{\sigma(\phi, \dot{s})(a')^\beta}{Z_\phi}$$

$$\mathrm{X}^\beta(\pi')P(\pi' \to \pi) \propto 2^{\beta\chi(\pi')} \cdot \sigma^\beta(\phi, \dot{s})(a) = 2^{\beta\chi(\pi')} \cdot \frac{\sigma(\phi, \dot{s})(a)^\beta}{Z_\phi}$$

The condition holds if $2^{\beta\chi(\pi)}\sigma(\phi, \dot{s})(a')^\beta = 2^{\beta\chi(\pi')}\sigma(\phi, \dot{s})(a)^\beta$, which is equivalent to $\frac{2^{\beta\chi(\pi)}}{2^{\beta\chi(\pi')}} = \frac{\sigma(\phi,\dot{s})(a)^\beta}{\sigma(\phi,\dot{s})(a')^\beta}$. We have just shown this identity to be true.

Thus, $\mathrm{X}^\beta$ is the stationary distribution. Since the chain is ergodic, it has a unique stationary distribution, so the time-averaged distribution of samples $\pi_r$ converges to $\mathrm{X}^\beta$. □

By running Gibbs sampling with large $\beta$, we are now able to find high-coherence d-policies.

## B.2 The Training-Friendly Variant

In practice, Gibbs sampling is easily applicable on in-context learning systems and Bayesian inference systems. However, each of its $N \cdot |\mathcal{S}|$ sampling steps samples from a different policy (obtained from leave-one-out datasets), and in a finetuning learning system, it would be a computational nightmare to do this many training runs.

If one is fine with approximations, then they could train one full policy $\sum_{n=1}^{|\mathcal{S}|} \pi_t(s_n)$ in each round, and before every sampling attempt, perform **unlearning** to erase one sample from such a policy.

Or, if the policy at hand can be easily merged (e.g. are ensemble models), then one may precompute prefix and suffix "sum policies", and merge one prefix and one suffix to obtain a leave-one-out policy. If storage is a concern, one could further apply square root decomposition.

However, the two workarounds are rather complicated and may not be applicable in many cases. Below is a more general method.

---

**Algorithm 2** Training-Friendly Variant of Gibbs Sampling

1: **Input:** Learning system $(\mathcal{M}, \mathcal{A}, \mathcal{S}, \sigma)$; initial d-policy $\pi_0$; number of steps $N$; temperature reciprocal $\beta$; **training split portion** $\gamma \in (0, 1)$.
2: **Output:** List of d-policies $\pi_0, \pi_1, \cdots, \pi_N$.
3: **procedure** TRAININGFRIENDLYGIBBS
4:     **for** $t = 0, 1, \ldots, N - 1$ **do**
5:         Independently and equiprobably select a $\lfloor \gamma |\mathcal{S}| \rfloor$-sized subset of $\mathcal{S}$, denoted with $\hat{\mathcal{S}}_t$.
6:         Let $\phi_t = \sum_{s \in \hat{\mathcal{S}}_t} \pi_t(s)$. **(Training step)**
7:         Independently sample $a_t^n \sim \sigma^\beta(\phi_t, s_n)$, for $s_n \notin \hat{\mathcal{S}}_t$. **(Sampling step)**
8:         Set $\pi_{t+1}$ such that $\pi_{t+1}(s_n) = a_t^n$ if $s_n \notin \hat{\mathcal{S}}_t$ and $\pi_{t+1}(s_n) = \pi_t(s_n)$ if $s_n \in \hat{\mathcal{S}}_t$.
9:     **end for**
10: **end procedure**

---

This algorithm is first implemented in an forthcoming, unpublished version of Wen et al. (2025). Note that it needs to be distinguished from what is typically called "iterative training", as the training step in each round always starts from zero, rather than from the previous step's trained policy $\phi_{t-1}$. Iterative training usually results in "model collapse" (Shumailov et al., 2024).

This algorithm approximates Gibbs sampling by replacing the computationally expensive leave-one-out context with a more economical "leave-a-random-chunk-out" context. In true Gibbs sampling, to resample the behavior for $s_n$, we use the context $\phi'_n = \sum_{m \neq n} \pi_t(s_m)$. In this training-friendly variant, we sample $a_t^n$ for all $s_n \notin \hat{\mathcal{S}}_t$ using the shared context $\phi_t = \sum_{s \in \hat{\mathcal{S}}_t} \pi_t(s)$. The set $\hat{\mathcal{S}}_t$ is a large, random subset of $\mathcal{S}$ of size $\lfloor \gamma |\mathcal{S}| \rfloor$.

The approximation holds if $\sigma(\phi_t, s_n) \approx \sigma(\phi'_n, s_n)$ for $s_n \notin \hat{\mathcal{S}}_t$. The contexts differ by the elements in $(\mathcal{S} \setminus \{s_n\}) \setminus \hat{\mathcal{S}}_t$. This set difference has size $(|\mathcal{S}| - 1) - \lfloor \gamma |\mathcal{S}| \rfloor \approx (1 - \gamma)|\mathcal{S}| - 1$. If the fraction of held-out behaviors $\gamma$ is close to 1, and the total number of contexts $|\mathcal{S}|$ is large, then $\phi_t$ is a very good approximation of $\phi'_n$. The logic is similar to the ICM approximation. In a system with rich, distributed information, the predictive distribution is robust to small perturbations in its context. Removing a small, random fraction of context behaviors is unlikely to drastically change the inference for another behavior.

Therefore, each sampling step in this algorithm is an approximation of a true Gibbs sampling step. While not exact, the process introduces a similar dynamic that pushes the d-policy towards regions of higher coherence, and its stationary distribution should be close to the target $X^\beta$, especially for $\gamma \to 1$.

Notably, this method can be seen as an extension of the classical (and surprising) training method of **pseudo-labeling** (Lee et al., 2013).

### B.3 Separating Consistent Personas from a Mixture

We test whether Gibbs sampling induces mode "snap" (convergence to a single coherent persona) instead of mode interpolation, mixture, or divergence. This is a lightly cherry-picked setup where the phenomenon is most salient; we tested two other setups whose results are directionally aligned but less consistent.

**Setup.** Note that this experiment measures stylistic convergence, not answer correctness, so benchmark contamination and saturation (e.g., in GSM8K-style questions) do not affect the conclusions. The experiment consists of 16 questions, including 12 scientific (e.g., "How does photosynthesis work?") and 4 emotional (e.g., "I just lost my job. . . "). We run 2048 rounds of Gibbs sampling ($2048 \times 16 = 32768$ inferences) with Claude-3.5-Haiku. Figure 7(a) displays embeddings for all generated texts over the rounds; circles for scientific questions, triangles for emotional questions; red for later rounds, blue for earlier rounds. Panels (b)–(d) show cosine similarity trends within scientific questions, within emotional questions, and between the two types.

**Results.** The observations are as follows: (i) scientific answers converge towards emotional answers while the latter stay put; (ii) within-type similarity first rises and then stabilizes; (iii) cross-type similarity continually rises. The implication is that the scientific mode is "snapping to" the emotional mode. The model does indeed answer scientific questions in emotional style after convergence, producing outputs like "A sacred invitation to explore the remarkable, infinite creativity of biochemical becoming" when asked the photosynthesis question. This demonstrates coherence optimization's tendency to converge on consistent personas rather than maintaining or amplifying diversity.

## C  Feedback-Free Self-Improvement Methods Reducible to Coherence Optimization

This section provides detailed arguments and proofs for the results showing that several state-of-the-art feedback-free self-improvement methods are reducible to coherence optimization.

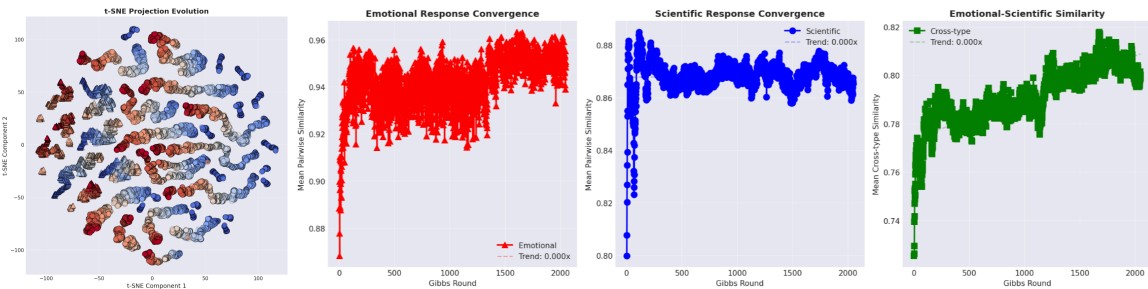

Figure 7: (a) Embeddings of scientific (circles) and emotional (triangles) answers over 2048 rounds of Gibbs sampling. Later rounds are colored red, earlier rounds blue. The model shifts to answering scientific questions in an emotional style. (b)(c)(d) Within-type and cross-type cosine similarity over 2048 rounds.

## C.1 Debate

We provide the detailed argument for Proposition 4.4. Consider a learning system with two contexts, $\mathcal{S} = \{s_{\mathrm{pro}}, s_{\mathrm{con}}\}$. $s_{\mathrm{pro}}$ is the set of possible answers to "Why is $P$ correct?" and $s_{\mathrm{con}}$ the set of answers to "Why is $P$ incorrect?", for a proposition $P$.

If we generate arguments independently via greedy decoding, the resulting d-policy $\pi_{\mathrm{greedy}} = (a_{\mathrm{pro}}, a_{\mathrm{con}})$ is likely to be incoherent. The proponent argument $a_{\mathrm{pro}}$ might make a claim that $a_{\mathrm{con}}$ trivially refutes, because $a_{\mathrm{pro}}$ did not anticipate this rebuttal. Generating $a_{\mathrm{con}}$ conditional on $a_{\mathrm{pro}}$ addresses only one direction of this problem.

A more robust conclusion arises from finding the d-policy $\pi^* = (a_{\mathrm{pro}}^*, a_{\mathrm{con}}^*)$ that maximizes the *joint* coherence $\chi(\pi)$. At zero temperature ($\beta = +\infty$), this is the d-policy found by infinite-beam beam search. Such a d-policy represents a "reflective equilibrium" between the two positions, where $a_{\mathrm{pro}}^*$ anticipates and counters $a_{\mathrm{con}}^*$, and vice versa.

This optimal d-policy can be found (or, for $\beta < \infty$, sampled from) using an iterative process that is formally identical to debate:

---

**Algorithm 3** Iterative Debate (Gibbs Sampling for $|\mathcal{S}| = 2$)

---

1: **Input:** Learning system $(\mathcal{M}, \mathcal{A}, \mathcal{S} = \{s_{\mathrm{pro}}, s_{\mathrm{con}}\}, \sigma)$; number of turns $N$; temperature reciprocal $\beta$.
2: **Output:** List of d-policies $\pi_1, \ldots, \pi_N$.
3: **procedure** DEBATE
4:      Independently sample $a_{\mathrm{pro}}^{(0)} \sim \sigma^\beta(0, s_{\mathrm{pro}})$                         ▷ Proponent makes first arg
5:      **for** $t = 0, 1, \ldots, N-1$ **do**
6:          Independently sample $a_{\mathrm{con}}^{(t+1)} \sim \sigma^\beta(a_{\mathrm{pro}}^{(t)}, s_{\mathrm{con}})$.      ▷ Opponent responds to proponent's last arg
7:          Independently sample $a_{\mathrm{pro}}^{(t+1)} \sim \sigma^\beta(a_{\mathrm{con}}^{(t)}, s_{\mathrm{pro}})$.      ▷ Proponent responds to opponent's last arg
8:          Set $\pi_{t+1} = (a_{\mathrm{pro}}^{(t+1)}, a_{\mathrm{con}}^{(t+1)})$.
9:      **end for**
10: **end procedure**

---

This iterative refinement, where each side updates its argument based on the other's, is the essence of both debate and Algorithm 1. The convergence result (Proposition 4.4) follows directly from Theorem 4.2, noting that Algorithm 3 is a synchronous variant where each party responds to the other's latest message rather than the second-to-latest; the proof of Theorem 4.2 is agnostic to this difference.

This debate setup, which optimizes coherence over a pair of conflicting contexts, illustrates a more general principle. Ideally, we would optimize coherence over all contexts $\mathcal{S}$ simultaneously (all contexts in the world, not merely two), especially with a strong reasoning model as the prior policy. The goal would be to find a

maximally consistent system of beliefs and behaviors, a form of reflective equilibrium. This is exactly the objective of Gibbs sampling.

## C.2 Simple Bootstrap Near $\beta = 1$

---
**Algorithm 4** Simple Bootstrap
---
1: **Input:** Learning system $(\mathcal{M}, \mathcal{A}, \mathcal{S}, \sigma)$; sequence of contexts $\{s_n\}_{n=1}^N$; temperature reciprocal $\beta$.
2: **Output:** Mapping $f : s_n \mapsto a_n \in s_n$.
3: **procedure** SIMPLEBOOTSTRAP
4:     **for** $n = 1, \ldots, N$ **do**
5:         Sample $a_n \sim \sigma^\beta \left( \sum_{m<n} a_m, s_n \right)$.
6:     **end for**
7: **end procedure**

---

We provide the proof of Proposition 4.5. The proof relies on the following technical assumption.

**Assumption C.1** (Context Independence on Random Subsets). A learning system exhibits quantitative context independence if for every $s \in \mathcal{S}$, there exists a limiting distribution $\rho_s$ and constants $C, K > 0, \alpha > 1$ such that for a randomly chosen sequence $(s_1, \ldots, s_{n-1})$ where $n \geq K$:

$$\mathbb{E}_{s_1, \ldots, s_{n-1}} \left[ D_{\mathrm{TV}} \left( \sigma \left( \sum_{m=1}^{n-1} a_m, s_n \right), \rho_{s_n} \right) \right] \leq \frac{C}{(n-1)^\alpha}$$

for any path $(a_1, \ldots, a_{n-1})$ and any $s_n \notin \{s_1, \ldots, s_{n-1}\}$.

*Proof of Proposition 4.5.* We first define the two distributions to be compared. The probability of generating a d-policy $\pi = (a_1, \ldots, a_N)$ via the sequential Simple Bootstrap process is:

$$P_{\mathrm{SB}}(\pi; \beta) = \prod_{n=1}^N \sigma^\beta \left( \sum_{m<n} a_m, s_n \right)(a_n) = \frac{\prod_{n=1}^N \sigma \left( \sum_{m<n} a_m, s_n \right)(a_n)^\beta}{\prod_{n=1}^N Z_n \left( \sum_{m<n} a_m \right)}$$

where $Z_n(\phi) = \sum_{a' \in s_n} \sigma(\phi, s_n)(a')^\beta$ is the local normalization constant at step $n$.

The target softmax-over-coherence distribution is $\mathrm{X}^\beta(\pi) = \frac{1}{Z} 2^{\beta \chi(\pi)}$. Using the definition $\chi(\pi) = \sum_{n=1}^N \log_2 \sigma \left( \sum_{m<n} a_m, s_n \right)(a_n)$, this becomes:

$$\mathrm{X}^\beta(\pi) = \frac{1}{Z} \prod_{n=1}^N \sigma \left( \sum_{m<n} a_m, s_n \right)(a_n)^\beta$$

By comparing these expressions, we find that $P_{\mathrm{SB}}(\pi; \beta)$ is a reweighting of $\mathrm{X}^\beta(\pi)$:

$$P_{\mathrm{SB}}(\pi; \beta) = \mathrm{X}^\beta(\pi) \cdot w(\pi), \quad \text{where} \quad w(\pi) = \frac{Z}{V(\pi)} \quad \text{and} \quad V(\pi) = \prod_{n=1}^N Z_n \left( \sum_{m<n} a_m \right)$$

Since $\sum_\pi P_{\mathrm{SB}}(\pi) = 1$, taking the expectation over $\mathrm{X}^\beta$ shows that $\mathbb{E}_{\mathrm{X}^\beta}[w(\pi)] = 1$. The total variation distance is given by:

$$D_{\mathrm{TV}}\left( P_{\mathrm{SB}}, \mathrm{X}^\beta \right) = \frac{1}{2} \mathbb{E}_{\mathrm{X}^\beta}\left[ |w(\pi) - 1| \right]$$

By Jensen's inequality, this is bounded by the standard deviation of the reweighting factor:

$$D_{\mathrm{TV}}^2 \leq \frac{1}{4} \mathbb{E}_{\mathrm{X}^\beta}\left[ (w(\pi) - 1)^2 \right] = \frac{1}{4} \mathrm{Var}_{\mathrm{X}^\beta}(w(\pi))$$

For a random variable $W$ concentrated around its mean, its relative variance is well-approximated by the variance of its logarithm: $\mathrm{Var}\left(W\right)/\mathbb{E}\left[W\right]^2 \approx \mathrm{Var}\left(\ln W\right)$. Applying this, we find $\mathrm{Var}\left(w\left(\pi\right)\right) \approx \mathrm{Var}\left(\ln V\left(\pi\right)\right)$. Our goal is to bound the variance of $\ln V\left(\pi\right)$.

**Case 1:** $\beta = 1$. In this case, $Z_n\left(\phi\right) = \sum_{a' \in s_n} \sigma\left(\phi, s_n\right)\left(a'\right) = 1$ for all $n$ and $\phi$. Thus, $V\left(\pi\right) = 1$ for all $\pi$. The relation becomes $P_{\mathrm{SB}}\left(\pi; 1\right) = \mathrm{X}^1\left(\pi\right) \cdot Z$. Since both distributions must sum to 1, we must have $Z = 1$. Therefore, for $\beta = 1$, the distributions are identical, i.e., $P_{\mathrm{SB}}\left(\pi; 1\right) = \mathrm{X}^1\left(\pi\right)$.

**Case 2:** $\beta$ **near 1.** Let $\beta = 1 + \epsilon$ for small $\epsilon$. A first-order expansion of $\ln Z_n\left(\phi\right)$ yields:

$$
\begin{aligned}
\ln Z_n\left(\phi\right) &= \ln \sum_{a'} \sigma\left(\phi, s_n\right)\left(a'\right)^{1+\epsilon} \\
&= \ln \sum_{a'} \sigma\left(\phi, s_n\right)\left(a'\right)\left(1 + \epsilon \ln \sigma\left(\phi, s_n\right)\left(a'\right) + O\left(\epsilon^2\right)\right) \\
&= \ln \left(1 + \epsilon \sum_{a'} \sigma\left(\phi, s_n\right)\left(a'\right) \ln \sigma\left(\phi, s_n\right)\left(a'\right) + O\left(\epsilon^2\right)\right) \\
&= -\epsilon H\left(\sigma\left(\phi, s_n\right)\right) + O\left(\epsilon^2\right)
\end{aligned}
$$

where $H\left(\cdot\right)$ is the Shannon entropy. The logarithm of the reweighting factor is therefore $\ln V\left(\pi\right) \approx -\epsilon S_H\left(\pi\right)$, where $S_H\left(\pi\right) = \sum_{n=1}^{N} H\left(\sigma\left(\sum_{m<n} a_m, s_n\right)\right)$. The variance is $\mathrm{Var}\left(\ln V\left(\pi\right)\right) \approx \epsilon^2 \mathrm{Var}\left(S_H\left(\pi\right)\right)$. We now show that $\mathrm{Var}\left(S_H\left(\pi\right)\right)$ is bounded by a constant independent of $N$.

We split the sum into a "burn-in" period ($n \leq K$) and a "stable" period ($n > K$).

In the **Stable Period** ($n > K$), the variance of this part of the sum has two components. First, we analyze the sum of the limit entropies, $\sum_{n=K+1}^{N} H\left(\rho_{s_n}\right)$. Since the sequence is a random permutation, this is a sum of $N - K$ values drawn without replacement from the finite population $\{H\left(\rho_s\right)\}_{s \in \mathcal{S}}$. Its variance is $\left(N - K\right)\sigma_H^2 \frac{N-(N-K)}{N-1} = \frac{K(N-K)}{N-1}\sigma_H^2$, where $\sigma_H^2$ is the population variance of the limit entropies. This variance is $O\left(K\right)$ and does not grow with $N$. Second, we analyze the sum of the deviations, $\sum_{n=K+1}^{N} \delta_n$, where $\delta_n = H\left(\sigma_n\right) - H\left(\rho_{s_n}\right)$. The variance of this sum is bounded by the sum of the individual variances (assuming weak correlation). The variance $\mathrm{Var}\left(\delta_n\right)$ is bounded by $\mathbb{E}\left[\delta_n^2\right]$, which by our assumption decays at least as fast as $O\left(n^{-\alpha}\right)$. Since $\alpha > 1$, the sum $\sum_{n=K+1}^{N} \mathrm{Var}\left(\delta_n\right)$ converges to a constant as $N \to \infty$. The rate of convergence is determined by the tail of the series, which is $\sum_{n=N+1}^{\infty} n^{-\alpha} = O\left(N^{1-\alpha}\right)$. The total variance of the stable period sum is thus bounded by a constant independent of $N$.

In the **Burn-in Period** ($n \leq K$), the first $K$ terms, $\sum_{n=1}^{K} H\left(\sigma\left(\sum_{m<n} a_m, s_n\right)\right)$, depend on the specific initial path and the specific initial $K$ contexts. As entropy is a bounded function, the variance of this finite sum is bounded by a constant $\Delta_K^2$ that depends on $K$ but not on $N$.

The total variance of $S_H\left(\pi\right)$ is therefore bounded by a constant $\Delta^2$ that is asymptotically independent of $N$. Combining our results:

$$
D_{\mathrm{TV}} \leq \frac{1}{2}\sqrt{\mathrm{Var}\left(w\left(\pi\right)\right)} \approx \frac{1}{2}\sqrt{\mathrm{Var}\left(\ln V\left(\pi\right)\right)} \approx \frac{1}{2}\sqrt{\epsilon^2 \mathrm{Var}\left(S_H\left(\pi\right)\right)} \leq \frac{1}{2}\left|\epsilon\right|\Delta
$$

Since $\epsilon = \beta - 1$ and $\Delta$ is a constant for a given system, we have shown that $D_{\mathrm{TV}}\left(P_{\mathrm{SB}}, \mathrm{X}^\beta\right) = O\left(\left|\beta - 1\right|\right)$. $\quad\square$

Note that the $O(\left|\beta - 1\right|)$ asymptotics limit the usefulness of the method. The most effective use of Gibbs sampling is when $\beta \gg 1$ and the converged-upon distribution is concentrated on the highest-coherence d-policies, which would not be well-approximated by simple bootstrap.

## C.3 Internal Coherence Maximization

We provide further analysis of the relationship between coherence and mutual predictability (Equations 2 and 3).

---

**Algorithm 5** Internal Coherence Maximization (Wen et al., 2025)

1: **Input:** Learning system $(\mathcal{M}, \mathcal{A}, \mathcal{S}, \sigma)$.
2: **Output:** d-policy $\pi^*$.
3: **procedure** ICM
4:     For d-policy $\pi$, define its *mutual predictability* $f_{\mathrm{MP}}(\pi) := \sum_{n=1}^{|\mathcal{S}|} \log_2 \sigma \left( \sum_{m \neq n} \pi(s_m), s_n \right) (\pi(s_n))$.
5:     Using discrete optimization methods, find $\pi^* = \arg\max_\pi f_{\mathrm{MP}}(\pi)$.
6: **end procedure**

---

The two objectives are approximately aligned under conditions where the learning system exhibits a form of informational saturation. The difference between the two objectives is:

$$f_{\mathrm{MP}}(\pi) - \chi(\pi) = \sum_{n=1}^{|\mathcal{S}|} \left[ \log_2 \sigma \left( \sum_{m<n} \pi(s_m) + \sum_{m>n} \pi(s_m), s_n \right) - \log_2 \sigma \left( \sum_{m<n} \pi(s_m), s_n \right) \right]$$

This term measures the total change in log-probability of the behaviors $\pi(s_n)$ when informed by "future" behaviors ($\sum_{m>n} \pi(s_m)$) in addition to "past" ones.

In systems with many weakly-dependent components (large $|\mathcal{S}|$), the context for predicting any single component becomes *informationally saturated* quickly. We say that a learning system exhibits informational saturation when, for large enough context $\sum_{m<n} \pi(s_m)$, the predictive distribution $\sigma(\sum_{m<n} \pi(s_m), s_n)$ changes negligibly upon adding further context $\sum_{m>n} \pi(s_m)$. When this holds, the difference term above is small, and the d-policy $\pi$ that maximizes one objective will be close to the one that maximizes the other.

This approximation is analogous to how pseudo-likelihood (based on local conditional probabilities, similar to $f_{\mathrm{MP}}$) is used as a computationally cheaper proxy for the true joint likelihood (similar to $\chi$) in fields like statistical physics and graphical models.

# D   Coherence Optimization as Policy-Wide Description Length Regularization

This section provides detailed proofs for the theoretical results in §5. We show that maximizing coherence imposes a policy-level regularization that improves out-of-sample generalization. As discussed in Appendix A, coherence can be further customized to regularize for high legibility, high factuality, or high alignment by appropriate choice of prior policy.

## D.1   Negated Coherence as Description Length

The concept of coherence has a natural and powerful interpretation, specifically the principle of Minimum Description Length (MDL). The MDL principle states that the best model for a set of data is the one that permits the shortest description of the data.

Consider a d-policy $\pi$ as a dataset, where each $\pi(s_n)$ is a data point. The coherence $\chi(\pi)$ is defined as a sum of sequential log-probabilities:

$$\chi(\pi) = \log_2 \sigma(0, s_1)(\pi(s_1)) + \log_2 \sigma(\pi(s_1), s_2)(\pi(s_2)) + \ldots$$

This is precisely the log-probability of observing the sequence of behaviors $(\pi(s_1), \pi(s_2), \ldots)$ under the predictive model defined by the learning system. By the chain rule, this is equal to the log of the joint probability, $\log_2 P(\pi(s_1), \ldots, \pi(s_{|\mathcal{S}|}))$.

According to Shannon's source coding theorem, the minimal average number of bits required to encode a message from a source is equal to its entropy. The optimal codelength for a specific message $x$ with probability $P(x)$ is $-\log_2 P(x)$ bits. Therefore, the negated coherence is the ideal codelength for the entire d-policy $\pi$:

$$\mathrm{CodeLength}(\pi) = -\log_2 P(\pi) = -\chi(\pi)$$

Maximizing coherence $\chi(\pi)$ is thus equivalent to minimizing the description length of the d-policy $\pi$. A d-policy is "coherent" if its constituent behaviors are highly predictable from one another, allowing for a very compact description. An "incoherent" policy consists of surprising or contradictory behaviors, requiring a longer description.

Coherence optimization, therefore, acts as a form of regularization that favors d-policies embodying strong, compressible patterns, and forces the model to discover and adhere to underlying principles rather than memorizing a set of unrelated behaviors.

### D.2 Uniform Convergence Theorem for Coherence

**Definition D.1** (Agreement and Accuracy). For any two d-policies $\pi_1, \pi_2 \in \prod_{s \in \mathcal{S}} s$ and a subset of contexts $\hat{\mathcal{S}} \subseteq \mathcal{S}$, we define the *agreement* between $\pi_1$ and $\pi_2$ on $\hat{\mathcal{S}}$ as

$$\alpha_{\hat{\mathcal{S}}}(\pi_1; \pi_2) = \alpha_{\hat{\mathcal{S}}}(\pi_2; \pi_1) := \frac{1}{|\hat{\mathcal{S}}|} \sum_{s \in \mathcal{S}} \mathbf{1}_{\pi_1(s) = \pi_2(s)}.$$

In particular, we denote $\alpha(\pi_1; \pi_2) := \alpha_{\mathcal{S}}(\pi_1; \pi_2)$. Given any *ground-truth d-policy* $\pi^*$, we define d-policy $\pi$'s *accuracy* on $\hat{\mathcal{S}}$ as $\alpha_{\hat{\mathcal{S}}}(\pi; \pi^*)$.

**Theorem D.2** (Uniform Convergence). *Let $\pi^*$ be any given ground-truth d-policy, and the* training contexts *$\hat{s}_1, \hat{s}_2, \cdots, \hat{s}_N$ be uniformly and independently sampled contexts from $\mathcal{S}$, for some $N > 0$. Let $\pi$ be an arbitrary d-policy with $\alpha_{\text{train}}(\pi; \pi^*) := \alpha_{\{\hat{s}_i\}_{i=1}^N}(\pi; \pi^*)$, then*

$$|\alpha(\pi; \pi^*) - \alpha_{\text{train}}(\pi; \pi^*)| \leq \sqrt{\frac{-2\chi(\pi) + \log_2 e + \log_2 \delta^{-1}}{2N}}$$

*holds uniformly for all $\pi$ with probability $1 - \delta$, for any given $\delta \in (0, 1)$.*

*Proof Sketch.* The proof follows the structure of PAC-Bayesian analysis. We define a prior distribution over the space of d-policies $P(\pi) \propto 2^{\chi(\pi)}$. A standard result in learning theory (a variant of the PAC-Bayes theorem) bounds the deviation between the true risk and the empirical risk. For any specific d-policy $\pi$, we can consider a "posterior" distribution that places all its mass on $\pi$. The KL divergence between this posterior and the prior is then $-\log_2 P(\pi) \propto -\chi(\pi)$. Applying the relevant concentration inequality yields a bound of the desired form, where the complexity term penalizing generalization is the negative log-prior, i.e., the negated coherence. $\square$

**Corollary D.3** (Coherence Regularization). *Under the conditions of the previous theorem, for any given set $\mathcal{R} \subseteq \prod_{s \in \mathcal{S}} s$ of d-policies and*

$$\hat{\pi} := \arg\max_{\pi \in \mathcal{R}} \left\{ \alpha_{\text{train}}(\pi; \pi^*) - \sqrt{\frac{-2\chi(\pi) + \log_2 e + \log_2 \delta^{-1}}{2N}} \right\} \tag{9}$$

$$\bar{\pi} := \arg\max_{\pi \in \mathcal{R}} \alpha(\pi; \pi^*), \tag{10}$$

*we have*

$$\alpha(\hat{\pi}; \pi^*) \geq \alpha(\bar{\pi}; \pi^*) - 2\sqrt{\frac{2\max(-\chi(\hat{\pi}), -\chi(\bar{\pi})) + \log_2 e + \log_2 \delta^{-1}}{2N}} \tag{11}$$

*with probability $1 - \delta$.*

In other words, by jointly maximizing training accuracy and coherence, we can make sure the resulting policy $\hat{\pi}$ is not much worse than the optimal policy $\bar{\pi}$ when tested out-of-sample.

*Remark* D.4 (Coherence Regularization Matters). One may be tempted to say "But LLMs are surely already trained on sufficiently many pretraining samples? Additional regularization matters little." This is not the case.

The effective dimensionality of the pretraining corpus is too small compared to the size of model parameters. The trillions of pretraining tokens are limited in effective dimension, because (i) natural language has lots of redundancies, and, more importantly, (ii) most human speeches on the Internet can be classified into a small number of discrete "personas" that explain a large portion of the variance in the opinions and styles of those speeches.

Empirically, we see that language models, even reasoning models, are much worse at creatively synthesizing novel ideas (e.g. hypothesizing and proving new theorems) than retrieving known ones from memory, while many humans can learn creative synthesis by consuming orders-of-magnitude less data. This is a sign that models overfit to statements of human knowledge by memorizing all of them, while being less successful at discovering generalizeable reasoning strategies that led to those knowledge in the first place.

Also empirically, we see that language models frequently express mutually contradicting information or opinions in different contexts, likely due to over- and under-representation of different information sources/viewpoints in different parts of the pretraining corpus. Let us zoom out and treat a (context, response persona) pair as a training sample, where each such sample encompasses hundreds of thousands of tokens, rather than (input token sequence, output next token) as a "sample". Under such a view, the number of training samples is drastically reduced; we would like the trained policy to generalize out-of-sample (i.e., to integrate information/opinions of all personas in all contexts) instead of overfitting to training samples (i.e., to always respond in the best-represented persona of the current context); and the fact that trained language models currently do the latter suggests that additional regularization is needed.

Coherence regularization is well-placed to address both examples of overfitting above, as (i) memorization by rote and (ii) emulating best-represented persona in every context both incur a multiplicative factor on the description length of the policy, and are thus strongly punished by coherence regularization. This sets coherence regularization apart from methods such as entropy regularization, which penalizes description length of policy behavior *on each input separately*. In contrast, coherence regularization controls the ***joint description length of all behaviors across all contexts***.

### D.3 Good and Bad Priors for Coherence Regularization

Let us look more closely at Equation 11. We can obviously replace the regularizer $\chi(\cdot)$ with $\chi_\phi(\cdot)$ for any arbitrary prior policy $\phi$, and the negative term on RHS will change accordingly. Since we want that term to be as close to 0 as possible, this gives us a measure of goodness for a prior policy.

**Definition D.5** (Optimality Gap). Given any ground-truth d-policy $\pi^*$, for any policy $\phi \in \mathcal{M}$, we define the *optimality gap* of $\phi$ as the prior policy to be

$$G(\phi; \pi^*) := -2\chi_\phi(\pi^*) + \log_2 e > 0$$

**Proposition D.6** (Optimality Gap Lower-Bounds Accuracy). *Under the conditions of the uniform convergence theorem, for any policy $\phi \in \mathcal{M}$, let*

$$\hat{\pi} := \operatorname*{arg\,max}_{\pi \in \prod_{s \in \mathcal{S}} s} \left\{ \alpha_{\text{train}}(\pi; \pi^*) - \sqrt{\frac{-2\chi_\phi(\pi) + \log_2 e + \log_2 \delta^{-1}}{2N}} \right\}, \tag{12}$$

*then, with probability $1 - \delta$,*

$$\alpha(\hat{\pi}; \pi^*) \geq 1 - \sqrt{\frac{2G(\phi; \pi^*) + 2\log_2 \delta^{-1}}{N}}.$$

*Proof.* By Equation 12, we have

$$\alpha_{\text{train}}(\hat{\pi}; \pi^*) - \sqrt{\frac{-2\chi_\phi(\hat{\pi}) + \log_2 e + \log_2 \delta^{-1}}{2N}} \geq \alpha_{\text{train}}(\pi^*; \pi^*) - \sqrt{\frac{-2\chi_\phi(\pi^*) + \log_2 e + \log_2 \delta^{-1}}{2N}}$$

$$= 1 - \sqrt{\frac{-2\chi_\phi(\pi^*) + \log_2 e + \log_2 \delta^{-1}}{2N}}$$

Thus,

$$\sqrt{\frac{-2\chi_\phi(\hat{\pi}) + \log_2 e + \log_2 \delta^{-1}}{2N}} \leq \sqrt{\frac{-2\chi_\phi(\pi^*) + \log_2 e + \log_2 \delta^{-1}}{2N}}$$

$$-\chi_\phi(\hat{\pi}) \leq -\chi_\phi(\pi^*)$$

Applying Corollary D.3 with $\mathcal{R} = \prod_{s \in \mathcal{S}} s$ completes the proof. $\qquad\square$

The propositions tells us that, the goodness of a prior policy is decided by the coherence of the optimal d-policy under such a prior. It's worth noting that **any policy can be a prior policy**, including e.g. a **(trainable) human preference model**. The job of coherence optimization is to turn any such prior policy (which, at inference time, can only evaluate the likelihood of *individual outputs/behaviors*) into a policy-level regularizer (which evaluates & regularizes the *joint* likelihood of the policy's behavior across contexts).

## D.4    Proofs for Asymptotic Optimality

This section provides the proof of the Regularization Optimality Theorem (Theorem 5.5), the Change-of-Prior Lemma (Lemma 5.6), and supporting material for §5.3.

*Optimality of Description-Length Regularization.* Under description-length regularization, with probability $1 - \delta$,

$$\alpha(\pi_{\text{SRM}}) \geq \max_{\pi \in \mathcal{A}^{\mathcal{S}}} \alpha(\pi) - 2\operatorname{reg}(\pi) \quad \text{(uniform convergence)}$$

$$\geq \mathrm{E}_{\pi \sim \mathcal{Q}}[\alpha(\pi) - 2\operatorname{reg}(\pi)]$$

$$\geq \mathrm{E}_{\pi \sim \mathcal{Q}}[\alpha(\pi)] - \sqrt{\frac{2}{N}} \, \mathrm{E}_{\pi \sim \mathcal{Q}}\left[\sqrt{\log_2 e/\delta - \log_2 \mathcal{P}(\pi)}\right].$$

Using the Taylor expansion, we get

$$\mathrm{E}_{\pi \sim \mathcal{Q}}\left[\sqrt{\log_2 e/\delta - \log_2 \mathcal{P}(\pi)}\right]$$

$$= \mathrm{E}_{\pi \sim \mathcal{Q}}\left[\sqrt{\log_2 e/\delta} - \frac{(1 + o(1))\log_2 \mathcal{P}(\pi)}{\sqrt{\log_2 e/\delta}}\right]$$

$$= \sqrt{\log_2 e/\delta} - \frac{(1 + o(1))}{\sqrt{\log_2 e/\delta}} \, \mathrm{E}_{\pi \sim \mathcal{Q}}\left[\log_2 \mathcal{P}(\pi)\right].$$

Substituting back:

$$\alpha(\pi_{\text{SRM}}) \geq \mathrm{E}_{\pi \sim \mathcal{Q}}[\alpha(\pi)] - \sqrt{\frac{2\log_2 e/\delta}{N}} + \sqrt{\frac{2 + o(1)}{N \log_2 e/\delta}} \, \mathrm{E}_{\pi \sim \mathcal{Q}}\left[\log_2 \mathcal{P}(\pi)\right].$$

Now, $\mathrm{E}_{\pi \sim \mathcal{Q}}[\log_2 \mathcal{P}(\pi)] = -\mathrm{H}[\mathcal{Q}] - \mathrm{KL}[\mathcal{Q}\|\mathcal{P}]$ by definition of cross-entropy. Simplifying:

$$\alpha(\pi_{\text{SRM}}) \geq \mathrm{E}_{\pi \sim \mathcal{Q}}[\alpha(\pi)] - \sqrt{\frac{2\log_2 e/\delta}{N}} + \sqrt{\frac{2 + o(1)}{N \log_2 e/\delta}} \, (\mathrm{H}\left[\mathcal{Q}\right] - \mathrm{KL}\left[\mathcal{Q} \,\|\, \mathcal{P}\right]).$$

$$\square$$

*Proof of Change-of-Prior Lemma.* This is a direct consequence of the definitions. We expand the terms. The left-hand side (LHS) is:

$$\text{LHS} = \sum_{l=1}^{L} \log_2 \sigma \left( \phi + \sum_{k=1}^{l-1} a_k^t, s_l^t \right) (a_l^t) + \sum_{n=1}^{N} \log_2 \sigma \left( r + \sum_{m=1}^{n-1} a_m^\phi, s_n^\phi \right) (a_n^\phi)$$

Substitute $\phi = r + \sum_{m=1}^{N} a_m^\phi$ into the first term:

$$\text{LHS} = \sum_{l=1}^{L} \log_2 \sigma \left( r + \sum_{m=1}^{N} a_m^\phi + \sum_{k=1}^{l-1} a_k^t, s_l^t \right) (a_l^t) + \sum_{n=1}^{N} \log_2 \sigma \left( r + \sum_{m=1}^{n-1} a_m^\phi, s_n^\phi \right) (a_n^\phi)$$

Now we expand the right-hand side (RHS), which is the coherence of the concatenated sequence of behaviors relative to the prior $r$. Denote the concatenated sequence as $\{a_i\}_{i=1}^{N+L}$, where $a_i = a_i^\phi$ for $i \le N$ and $a_i = a_{i-N}^t$ for $i > N$.

$$\text{RHS} = \hat{\chi}_r[a_1, \cdots, a_{N+L}] = \sum_{i=1}^{N+L} \log_2 \sigma \left( r + \sum_{j=1}^{i-1} a_j, s_i \right) (a_i)$$

$$= \sum_{i=1}^{N} \log_2 \sigma \left( r + \sum_{j=1}^{i-1} a_j^\phi, s_i^\phi \right) (a_i^\phi) + \sum_{i=N+1}^{N+L} \log_2 \sigma \left( r + \sum_{j=1}^{i-1} a_j, s_i \right) (a_i)$$

The first term of the RHS is exactly the second term of the LHS. Now let's examine the second term of the RHS. For $i = N + l$ (where $l \in [1, L]$), the context sum is $\sum_{j=1}^{i-1} a_j = \sum_{j=1}^{N} a_j^\phi + \sum_{j=N+1}^{N+l-1} a_j = \sum_{j=1}^{N} a_j^\phi + \sum_{k=1}^{l-1} a_k^t$. So the second term of the RHS is:

$$\sum_{l=1}^{L} \log_2 \sigma \left( r + \sum_{j=1}^{N} a_j^\phi + \sum_{k=1}^{l-1} a_k^t, s_l^t \right) (a_l^t)$$

This is exactly the first term of the LHS. Therefore, LHS = RHS. $\qquad\square$

---

**Example D.7** (Conditional Probabilities in Bayesian Learning System)**.** In a Bayesian learning system, Theorem 5.8 becomes a simple identity of log-probabilities. The base coherence $\chi(\pi)$ is the log joint probability of all behaviors in the d-policy $\pi$: $\chi(\pi) = \log_2 P(\pi(\mathcal{S})) = \log_2 P(\pi(\mathcal{S}_a), \pi(\mathcal{S}_b))$. The "pretrain coherence" $\hat{\chi}_0[\pi(\mathcal{S}_b)]$ is the log marginal probability of the pretraining behaviors: $\hat{\chi}_0[\pi(\mathcal{S}_b)] = \log_2 P(\pi(\mathcal{S}_b))$. The "posttrain coherence w.r.t. pretrain prior" $\chi^a(\pi(\mathcal{S}_a))$ corresponds to coherence in a quotient system where the base policy is $0_a = \sum_{s \in \mathcal{S}_b} \pi(s)$. This is equivalent to starting with a prior conditioned on the pretraining data $\pi(\mathcal{S}_b)$. Thus, this term is the log conditional probability: $\chi^a(\pi(\mathcal{S}_a)) = \log_2 P(\pi(\mathcal{S}_a)|\pi(\mathcal{S}_b))$. Plugging these into the first half of Equation 7:

$$\underbrace{\log_2 P(\pi(\mathcal{S}_a)|\pi(\mathcal{S}_b))}_{\chi^a(\pi(\mathcal{S}_a))} + \underbrace{\log_2 P(\pi(\mathcal{S}_b))}_{\hat{\chi}_0[\pi(\mathcal{S}_b)]} = \underbrace{\log_2 P(\pi(\mathcal{S}_a), \pi(\mathcal{S}_b))}_{\chi(\pi)}$$

This is simply the chain rule of probability, $\log_2 P(A|B) + \log_2 P(B) = \log_2 P(A, B)$. The other half of the equation follows from symmetry, as $P(A, B) = P(B, A) = P(B|A)P(A)$, which corresponds to the right-hand side of Equation 7.

---

## D.5 Heuristic Argument for the Equivalence Between Regularization and Optimization

We now informally show why the two formulations in Conjecture 5.9 are likely equivalent under appropriate conditions. By Equation 7, optimizing coherence on the posttrain samples $\pi(\mathcal{S}_a)$ with respect to a pretrained

policy is equivalent to jointly optimizing (i) the coherence of posttrain samples with respect to zero prior and (ii) the accuracy of a posttrain-only policy on pretrain samples. (Since $\pi(\mathcal{S}_b)$ is fixed, we may ignore the "pretrain coherence" term.)

This looks similar to the regularized training objective from Proposition 5.4, but there are two gaps to bridge. First, $\hat{\chi}_0\left[\pi(\mathcal{S}_a)\right]$ appears linearly in the coherence objective, while it appears under a square root in the regularization term $\sqrt{[-2\chi(\pi) + \log_2 e + \log_2(1/\delta)]/2N}$. Second, $\chi^b(\pi(\mathcal{S}_b))$ is coherence on pretrain samples, not accuracy $\alpha_{\text{train}}(\pi)$.

We address both gaps with heuristic calculations.

Let's first look at the second question. We know that in a naive Bayes model, the log likelihood scales linearly with accuracy times sample size. In supervised training, this is approximately true up to a constant factor. In LLM finetuning pipelines, the cross-entropy loss typically drops by around a factor of 2 before stagnating, so the cumulative loss is approximately linear up to a factor of 2. The same is true of pretraining loss after the initial stage of rapid loss decline. We may thus assume

$$\left| c|\mathcal{S}_b|(\alpha^b(\pi) - 1) - \chi^b(\pi(\mathcal{S}_b)) \right| \leq \epsilon|\mathcal{S}_b|$$

for some constants $c > 0, d < 0$ and small $\epsilon$. This answers the second question. After we multiply the pretrain accuracy by $c|\mathcal{S}_b|$, the two are approximately equal in size.

Now, the first question. Observe that Proposition 5.4 does not rely on the accuracy and coherence being calculated on the same context set, and not even that the two being calculated on the same number of contexts. We only require that *the set of contexts we use for specifying the trained d-policy be included in the calculation of coherence.* We can thus switch the $\chi(\pi)$ in the regularization term for $\hat{\chi}_0\left[\pi(\mathcal{S}_a)\right]$ (where $\mathcal{S}_a$ is the "set of contexts we use for specifying d-policy"), and $N$, on the other hand, for $|\mathcal{S}_b|$ (the number of samples used for training accuracy calculation). We must also multiply the regularization term by the same factor of $c|\mathcal{S}_b|$, so we are now comparing $\hat{\chi}_0\left[\pi(\mathcal{S}_a)\right]$ and

$$c|\mathcal{S}_b|\sqrt{\frac{-2\hat{\chi}_0\left[\pi(\mathcal{S}_a)\right] + \log_2 e + \log_2(1/\delta)}{2|\mathcal{S}_b|}} = c\sqrt{|\mathcal{S}_b|} \cdot \sqrt{-\hat{\chi}_0\left[\pi(\mathcal{S}_a)\right] + \frac{\log_2 e + \log_2(1/\delta)}{2}}$$

Now we have an oracle that, when given any $\mathcal{S}_a, \mathcal{S}_b, \pi(\mathcal{S}_b)$, can tell us the optimum for

$$\max_{\pi(\mathcal{S}_a)} \left( c|\mathcal{S}_b|\alpha^b(\pi) + \hat{\chi}_0\left[\pi(\mathcal{S}_a)\right] \right)$$

$$= \max_{r \in \mathbb{R}} \left( \max_{\pi(\mathcal{S}_a): \; -\hat{\chi}_0[\pi(\mathcal{S}_a)]/|\mathcal{S}_a|=r} c|\mathcal{S}_b|\alpha^b(\pi) - |\mathcal{S}_a| \cdot r \right)$$

$$:= \max_{r \in \mathbb{R}}(f(r) - |\mathcal{S}_a| \cdot r).$$

while we would like to solve for the regularized accuracy, i.e., roughly

$$\max_{\pi(\mathcal{S}_a)} \left( c|\mathcal{S}_b|\alpha^b(\pi) - c\sqrt{|\mathcal{S}_b|} \cdot \sqrt{-\hat{\chi}_0\left[\pi(\mathcal{S}_a)\right] + \frac{\log_2 e + \log_2(1/\delta)}{2}} \right)$$

$$= \max_{r \in \mathbb{R}} \left( \max_{\pi(\mathcal{S}_a): \; -\hat{\chi}_0[\pi(\mathcal{S}_a)]/|\mathcal{S}_a|=r} c|\mathcal{S}_b|\alpha^b(\pi) - c\sqrt{|\mathcal{S}_b|} \cdot \sqrt{|\mathcal{S}_a| \cdot r + \frac{\log_2 e + \log_2(1/\delta)}{2}} \right)$$

$$\approx \max_{r \in \mathbb{R}} \left( \max_{\pi(\mathcal{S}_a): \; -\hat{\chi}_0[\pi(\mathcal{S}_a)]/|\mathcal{S}_a|=r} c|\mathcal{S}_b|\alpha^b(\pi) - \sqrt{|\mathcal{S}_a|} \cdot c\sqrt{|\mathcal{S}_b|} \cdot \sqrt{r} \right)$$

$$:= \max_{r \in \mathbb{R}} \left( f(r) - \sqrt{|\mathcal{S}_a|} \cdot g(r) \right). \tag{13}$$

Note that, since $-\hat{\chi}_0\left[\pi(\mathcal{S}_a)\right]$ is approximately asymptotically linear with respect to $|\mathcal{S}_a|$, by defining $r$ with $-\hat{\chi}_0\left[\pi(\mathcal{S}_a)\right]/|\mathcal{S}_a|$, we ensure that the function $f(r)$ doesn't vary by orders of magnitude when $|\mathcal{S}_a|$ changes.

We will now assume $f(r)$ is a fixed function independent of $|\mathcal{S}_a|$, which is true when $|\mathcal{S}_a|$ is large and the ratio $r$ has converged for every possible d-policy.

$g(r)$ is concave. Empirically, $f(r)$ is also likely approximately concave, as it represents the tradeoff between coherence ($\approx$ accuracy) on $\mathcal{S}_a$ and accuracy on $\mathcal{S}_b$, and accuracy tradeoff on two different datasets likely exhibits diminishing returns.

Now, remember that there's one more variable we can control, but which we've so far not put to use: $|\mathcal{S}_a|$. We will now use it as a Lagrange multiplier, and denote it with $\lambda$.

Given any $\lambda$, the oracle gives us the optimal $r^* : f'(r^*) = \lambda$. Our hope is to find a value for $\lambda$ such that $r^*(\lambda)$ also satisfies the optimality condition for Equation 13, i.e.,

$$f'(r^*) - \sqrt{\lambda} \cdot g'(r^*) = \lambda - \sqrt{\lambda} \cdot g'(r^*) = \lambda - \sqrt{\lambda} \cdot \frac{1}{2}c\sqrt{\frac{|\mathcal{S}_b|}{r^*}} = 0 \iff \lambda = \frac{c^2}{4r^*}|\mathcal{S}_b|$$

Now we know that such a $|\mathcal{S}_a|^*$ exists and

$$
\begin{aligned}
|\mathcal{S}_a|^* &\approx \frac{c^2}{4r^*}|\mathcal{S}_b| \\
&\approx \frac{\left(-\chi^b(\pi^*(\mathcal{S}_b))/[|\mathcal{S}_b|(1-\alpha^b(\pi^*))]\right)^2}{4(-\hat{\chi}_0[\pi^*(\mathcal{S}_a)])/|\mathcal{S}_a|^*}|\mathcal{S}_b| \\
&= \frac{1}{4} \times \underbrace{\frac{\overbrace{\left(-\chi^b(\pi^*(\mathcal{S}_b))/|\mathcal{S}_b|\right)^2}^{\text{mean pretrain coherence, squared}}}{-\hat{\chi}_0[\pi^*(\mathcal{S}_a)]/|\mathcal{S}_a|^*}}_{\text{mean posttrain coherence, inversed}} \times \underbrace{\left(\frac{1}{1-\alpha^b(\pi^*)}\right)^2}_{\text{pretrain error rate, inverse-squared}} \times \underbrace{|\mathcal{S}_b|}_{\text{pretrain sample count}} .
\end{aligned}
$$

# E   Out-of-Sample Evaluation of Gibbs Sampling on GSM8K

We report out-of-sample (test) accuracy for the training-friendly Gibbs sampling experiment from §6. The original experiment only reported in-sample (train) accuracy. Here we use the same setup with an expanded evaluation. We run 100 seeds per $\gamma$ value, with 100 train questions and 200 held-out test questions from GSM8K. The model is `meta-llama/Llama-3.2-1B-Instruct`, the temperature is 0.3, and we run 40 Gibbs rounds for each $\gamma \in \{0.25, 0.45, 0.65, 0.85\}$. The test questions are never included in the Gibbs sampling context; they are only used for evaluation.

**Results.**   Table 2 reports the initial, final, and peak accuracies for both train and test splits. Figure 8 shows the train and test accuracy trajectories over Gibbs rounds.

We could make several observations. First, test accuracy improves substantially for all $\gamma$ values, rising from roughly 30% at initialization to 38–40% at the final round. The mean peak (best round of the seed-averaged trajectory) reaches 39–41%, while the oracle peak (average of per-seed maxima, where each seed selects its own best round) reaches 46–47%. The gap between these two numbers reflects the noisiness of individual runs. Both confirm that coherence optimization generalizes to held-out questions rather than merely memorizing the in-sample context.

Second, the train-test gap behaves differently across $\gamma$. At $\gamma = 0.25$ (small hold-out fraction, strongest resampling), the mean peak reaches 45.7% train vs. 39.3% test, a gap of about 6 percentage points. At $\gamma = 0.85$ (large hold-out fraction, weakest resampling), the gap essentially vanishes (40.7% train vs. 40.4% test). Higher $\gamma$ values sacrifice some in-sample performance but generalize more evenly, consistent with the regularization interpretation.

Third, while in-sample accuracy varies considerably with $\gamma$ (from 40.4% to 44.9% at round 40), test accuracy is remarkably stable across $\gamma$ values (from 38.1% to 40.2% at round 40, and from 39.3% to 41.2% at the mean peak). This suggests that the generalization benefit of coherence optimization is robust to the choice of $\gamma$, even though the in-sample signal is stronger at lower $\gamma$.

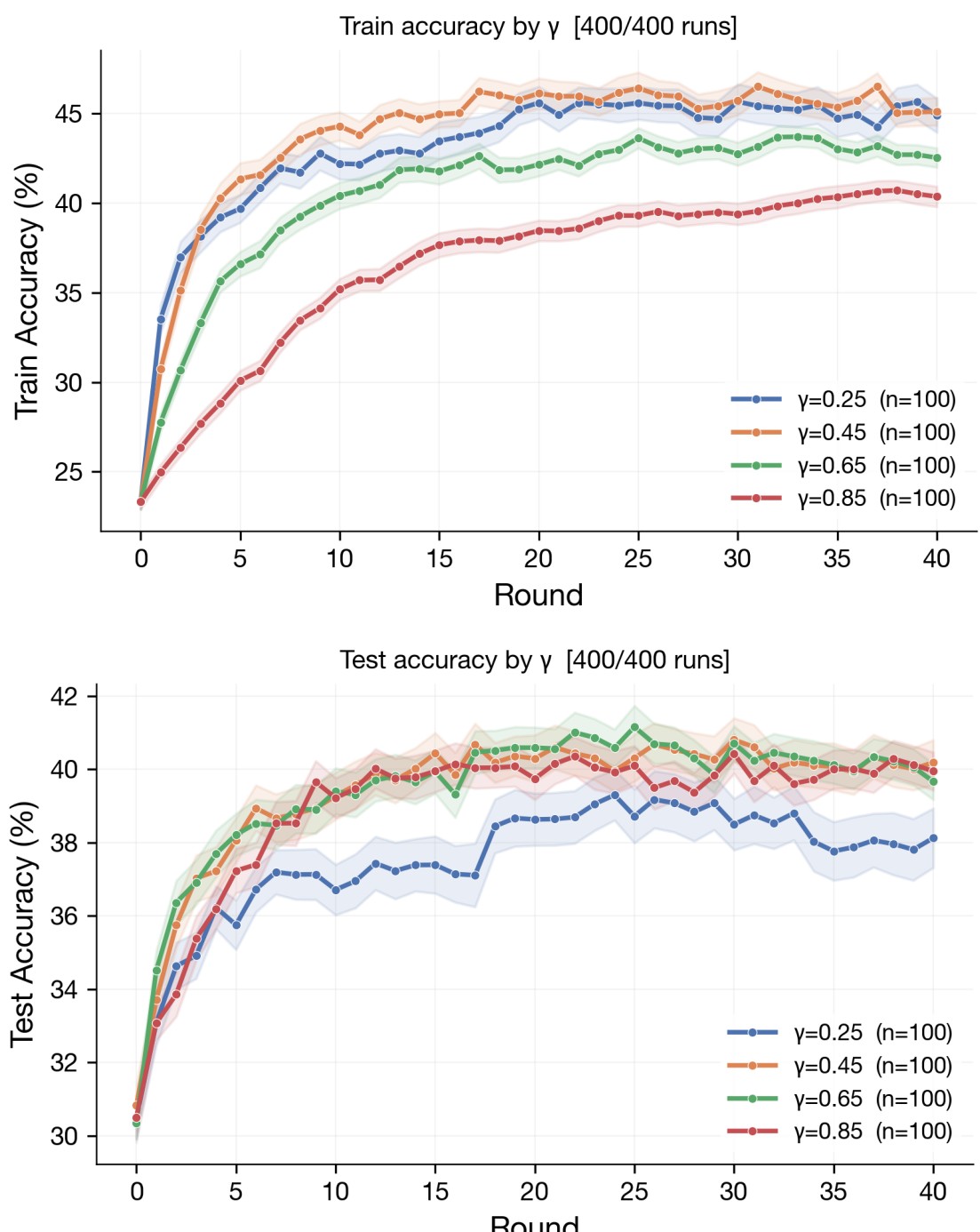

Figure 8: Train (top) and test (bottom) accuracy over 40 Gibbs rounds for $\gamma \in \{0.25, 0.45, 0.65, 0.85\}$, averaged over 100 seeds per $\gamma$ (shaded region: 95% CI).

Table 2: Train and test accuracy over 40 Gibbs rounds, averaged over 100 seeds per $\gamma$ ($\pm$ SE). We report two notions of peak accuracy. **Mean peak** first averages accuracy across seeds at each round, then takes the maximum of this averaged trajectory. **Oracle peak** takes the maximum accuracy across rounds for each seed individually, then averages these per-seed maxima. The oracle peak is higher because each seed selects its own best round.

| $\gamma$ | Init (%) | Final (%) | Mean peak (%) | Oracle peak (%) | Improv. (%) |
|---|---|---|---|---|---|
| *Train (100 questions, in-sample)* | | | | | |
| 0.25 | $23.3 \pm 0.4$ | $44.9 \pm 1.0$ | 45.7 (r30) | $56.2 \pm 0.6$ | $+21.6 \pm 1.0$ |
| 0.45 | $23.3 \pm 0.4$ | $45.1 \pm 0.8$ | 46.5 (r31) | $54.9 \pm 0.6$ | $+21.8 \pm 0.9$ |
| 0.65 | $23.3 \pm 0.4$ | $42.5 \pm 0.5$ | 43.7 (r33) | $50.6 \pm 0.5$ | $+19.2 \pm 0.6$ |
| 0.85 | $23.3 \pm 0.4$ | $40.4 \pm 0.6$ | 40.7 (r38) | $45.2 \pm 0.5$ | $+17.0 \pm 0.7$ |
| *Test (200 questions, out-of-sample)* | | | | | |
| 0.25 | $30.5 \pm 0.6$ | $38.1 \pm 0.8$ | 39.3 (r24) | $46.6 \pm 0.4$ | $+7.7 \pm 0.9$ |
| 0.45 | $30.8 \pm 0.6$ | $40.2 \pm 0.6$ | 40.8 (r30) | $47.0 \pm 0.4$ | $+9.4 \pm 0.8$ |
| 0.65 | $30.4 \pm 0.5$ | $39.7 \pm 0.5$ | 41.2 (r25) | $46.9 \pm 0.4$ | $+9.3 \pm 0.7$ |
| 0.85 | $30.5 \pm 0.6$ | $40.0 \pm 0.5$ | 40.4 (r30) | $46.1 \pm 0.4$ | $+9.5 \pm 0.8$ |

