# OpenReview forum: "Self-Improvement as Coherence Optimization: A Theoretical Account"
_TMLR — Accepted by TMLR_

### Review · Reviewer_BNBn · 2026-03-07

**Summary Of Contributions:**

This paper is a novel theoretical effort to understand and unify several recent self-improvement techniques originating from empirical work on Large Language Models (LLMs), such as debate and internal coherence maximization. The main theoretical framework proposed is based on a precise mathematical definition of coherence, or the joint likelihood of the model's behaviors across all contexts. The work makes several key contributions:
* Unifies a number of empirical self-improvement methods under the coherence optimization framework.
* Proves that coherence optimization is the optimal form of description-length regularization.
* Proposes a practical, theoretically grounded algorithm for general coherence optimization.

Key Strengths
* An attempt to provide theoretical analysis of recent empirical advances in self-improving large models. This is quite rare and represents a strong positive contribution.
* Definitions, derivations, and proofs are precise and largely correct (in the parts the reviewer was able to check, not an expert in the area).
* Efforts are made to connect the theory to practical applications via a new proposed algorithm and by casting existing methods such as in-context learning and fine-tuning in the coherence optimization theory.

Key Weaknesses
* The current state of writing and theory is dense, and likely unapproachable for empirical researchers. While this doesn't take away from the theoretical contributions of this work, it's hard to see how this would have much impact on empirical methods like debate.
* Experiments are, as the authors state, preliminary. GSM8K is a weak benchmark currently due to contamination and saturation. There are no baseline comparisons to any other self-improvement or training methods, even those mentioned in the paper.

**Audience:**

Yes

**Audience Explanation:**

Yes, but likely limited to the learning theory community due to the theoretical focus of this work and weak experimental section.

**Broader Impact Concerns:**

None.

**Claims And Evidence:**

Yes

**Claims Explanation:**

As a mainly theory paper, the claims are mostly supported by rigorous proofs which the paper provides. Best efforts were made to check the proofs, and the derivations are high quality. The reviewer is, however, not confident in the correctness of every line.

Some parts of the claim that coherence optimization is an umbrella unifying framework for existing techniques are a bit weak. For example, the arguments in Examples 3.4 and 3.5 showing how in-context learning and fine-tuning are approximations of coherence optimization. Both use very similar arguments that "they are 1st-order approximations of Bayesian inference" (a connection that is also a bit tenuous). Another example is the argument on context saturation at the top of page 32, arguing for the relationship between coherence and mutual predictability (e.g., what are "complex systems"?).

Preliminary experiment results do show some advantage for their proposed algorithm, but are less convincing due to the very few benchmarks and lack of realistic baselines.

**Requested Changes:**

[minor importance] Section 3.1: what is the definition of a behaviour? Of A? Similarly, in Definition 3.2, what is the definition of M?

[minor importance] It could just be the writing in the introduction, but it is not completely clear why description-length regularization is optimal for self-supervised learning. The closest justification is currently "One large class of regularizers are those constructed from description lengths" (in the 4th paragraph), but this does not directly prove optimality. This connection should be clearer and more self-contained in the paper.

[minor importance] Does the framework allow external information or not? There's an inconsistency between the introduction, which focuses on self-improvement without external supervision, and for instance, the answer to FAQ 1, which states the method is a "regularization technique that functions on top of supervised signals" and also discusses using RL, which requires an external reward signal.

[moderate importance] The answer to FAQ 4 ("Does coherence optimization require a fully truthful model to begin with?") is hand-wavy. What does "substantially truthful" mean exactly? FAQ 4 was also a question I had, but the answer suggests more is needed for coherence optimization to work?

---

> ### Author Response · Authors · 2026-04-08
>
> > - The current state of writing and theory is dense, and likely unapproachable for empirical researchers. While this doesn't take away from the theoretical contributions of this work, it's hard to see how this would have much impact on empirical methods like debate.
>
> We appreciate this feedback. We believe the theoretical contributions are needed precisely because the field of feedback-free self-improvement has been entirely empirical and heuristic, and a formal understanding of why these methods work can guide future empirical work. That said, we have added a "Practical implications" paragraph at the end of Section 5 that summarizes the key takeaways for practitioners in plain language. Namely, we distill three actionable points: (1) the prior should approximate the data-generating distribution as closely as possible, (2) the number of unsupervised contexts should be comparable to the effective number of supervised samples, and (3) overoptimization can be mitigated through early stopping, temperature tuning, or limiting the unsupervised context count.
>
> > - Experiments are, as the authors state, preliminary. GSM8K is a weak benchmark currently due to contamination and saturation. There are no baseline comparisons to any other self-improvement or training methods, even those mentioned in the paper.
>
> We agree the experiments are preliminary and have labeled them as such. Regarding GSM8K, the persona experiment (Section 6.2) measures stylistic convergence (mode snap), not answer correctness. Benchmark contamination and saturation therefore do not affect its conclusions. We have added a note in the experimental setup making this explicit. The HARDMath experiment (Section 6.1) uses a more challenging and less saturated benchmark. We have also expanded the Limitations section to discuss what a rigorous baseline comparison against other regularizers would require.
>
> > - Some parts of the claim that coherence optimization is an umbrella unifying framework for existing techniques are a bit weak. For example, the arguments in Examples 3.4 and 3.5 showing how in-context learning and fine-tuning are approximations of coherence optimization. Both use very similar arguments that "they are 1st-order approximations of Bayesian inference" (a connection that is also a bit tenuous). Another example is the argument on context saturation at the top of page 32, arguing for the relationship between coherence and mutual predictability (e.g., what are "complex systems"?).
>
> We want to clarify the structure of the argument. Examples 3.4 and 3.5 do *not* claim that ICL and finetuning are coherence optimization. They define ICL and finetuning as *learning systems*, which are formal abstractions that provide tractable priors. The actual unification claim is that methods like debate, bootstrap, and ICM optimize coherence *as defined through* these learning systems (Section 4.2). The "1st-order approximation of Bayesian inference" caveat is about how well the learning system abstraction fits reality, not about the coherence optimization claim itself. We have added clarifying sentences in Examples 3.4 and 3.5 to make this distinction explicit.
>
> Regarding context saturation (Appendix C.3), we have replaced "complex systems" with a more precise term ("systems with many weakly-dependent components") and added a definition of informational saturation. A learning system exhibits informational saturation when, for a large enough context, the predictive distribution changes negligibly upon adding further context.

---

> > ### Author Response · Authors · 2026-04-08
> >
> > > [minor importance] Section 3.1: what is the definition of a behaviour? Of A? Similarly, in Definition 3.2, what is the definition of M?
> >
> > $\mathcal{A}$ is the behavior space, defined at the start of Section 3.1 as "a finite set of all possible behaviors." Each behavior $a \in \mathcal{A}$ belongs to exactly one context $s \in \mathcal{S}$. The meaning of a behavior depends on the context; for example, when $s$ is a question or prompt, $a$ is a potential response to it. $\mathcal{M}$ is the policy monoid, defined in Definition 3.2. We have added a brief introduction of $\mathcal{M}$ in the paragraph preceding Definition 3.2, describing it as a space of stochastic policies (e.g., training datasets or in-context example sets) with an inference function $\sigma$.
> >
> > > [minor importance] It could just be the writing in the introduction, but it is not completely clear why description-length regularization is optimal for self-supervised learning. The closest justification is currently "One large class of regularizers are those constructed from description lengths" (in the 4th paragraph), but this does not directly prove optimality. This connection should be clearer and more self-contained in the paper.
> >
> > Thank you. The introduction's phrasing was indeed too compressed. The logical chain is as follows. Description-length regularizers form a classical and general family for controlling overfitting. Among all such regularizers, the one derived from a KL-optimal prior yields the tightest worst-case accuracy guarantee (Theorem 5.5). Coherence is the description length under the learning system's prior (Section 5.1). Therefore coherence regularization with a pretrained prior is optimal within this family. We have rewritten the relevant paragraph in the introduction to lay out this chain more clearly.
> >
> > > [minor importance] Does the framework allow external information or not? There's an inconsistency between the introduction, which focuses on self-improvement without external supervision, and for instance, the answer to FAQ 1, which states the method is a "regularization technique that functions on top of supervised signals" and also discusses using RL, which requires an external reward signal.
> >
> > We acknowledge the framing was unclear, but there is no true inconsistency. Coherence optimization always requires a pretrained prior, and producing that prior requires supervised data (the pretraining corpus). So when viewed as part of the full training pipeline, coherence optimization is semi-supervised. The supervised signal is embedded in the prior, and coherence optimization leverages it through regularization. When we zoom in on the coherence optimization process itself, it is purely unsupervised: given a fixed prior, it optimizes coherence without any additional labels. The introduction emphasizes the latter perspective because it is the most surprising aspect. FAQ 1 explains the former perspective. Both are correct descriptions at different levels of abstraction. We have added clarifying sentences in the introduction and in FAQ 1 to bridge between these two perspectives.
> >
> > > [moderate importance] The answer to FAQ 4 ("Does coherence optimization require a fully truthful model to begin with?") is hand-wavy. What does "substantially truthful" mean exactly? FAQ 4 was also a question I had, but the answer suggests more is needed for coherence optimization to work?
> >
> > We agree this needed a more precise answer. We have revised FAQ 4 with a formal definition. A prior is "substantially truthful" if the optimality gap of the ground-truth d-policy, $G(\phi; \pi^\*) = -2\chi_\phi(\pi^\*) + \log_2 e$, is smaller than the expected optimality gap $\mathbb{E}_{\pi'}[G(\phi; \pi')]$ for a random d-policy $\pi'$ drawn from the prior. In other words, truth must outperform the average response under the prior, though it need not be the most likely answer to every individual question. When this condition is violated, the generalization bound (Proposition 5.4) becomes vacuous, and coherence optimization may converge to a coherent-but-false equilibrium. This is exactly the failure mode the theory predicts. We also describe two practical strategies (interleaved training and weak-to-strong) for ensuring the condition holds.

---

### Review · Reviewer_LjMB · 2026-03-25

**Summary Of Contributions:**

The paper provide a general framework to understand several method that helps improve language model accuracy without external supervision. They define a deterministic policy for language tasks and show that optimizing its coherence is optimal for semi-supervised learning when the regularizer is derived from a retrained model. They showed that the KL optimal prior is the best available regularizer and Gibbs sampling efficiently maximize the coherence. Several successful methods like debate/bootstrap/ICM can be reduced to Gibbs Sampling.

**Audience:**

Yes

**Audience Explanation:**

This paper provides understanding to several popular empirical method for improving language models, such as debates.

**Broader Impact Concerns:**

It is sufficiently discussed.

**Claims And Evidence:**

Yes

**Claims Explanation:**

The learning systems is well set up and the theoretical results are well written and seems correct. The claims are also complemented with empirical evidence.

**Requested Changes:**

1. My biggest confusion and concern is with the chain rule assumption. Because maximize $P(\pi ^\ast)$ means the prior should assign high joint probability to the correct answer to every question simultaneously, you would need the chain rule to hold so that the retrained model is a valid instance of prior. If one instead define $D$ to be over the marginals, then coherence optimization seems to reduce to greedy decoding and the results are not interesting anymore. The satisfaction of chain rule is rather counter intuitive for language models, and as the paper correctly pointed out, it seems to be only fully satisfied by the Bayesian learning system (while the other two, arguably, are the learning systems in interest).  I wonder if you could provide an approximation error on the chain rule or something, to better quantify how much in context learning system and fine-tuning learning system satisfy this and to better justify coherence optimization?

2. The definition of coherence is on the probability of model assigned to exact outputs. But two words having the same semantics meanings are interpreted as two different exact output. Say if the model have learned $ P( yes \mid yeah) \approx P(yes \mid yes)  $, will coherence treat them similarly? If so, it might be worth to point this out and discuss some of the results under this.

3. One big motivation for coherence is for it to be a generalization regularizer. However, Table 1 only reports training accuracy and it seems like no out-of-sample evaluation is performed. Could you add reports on the test accuracy?


4. Another important claim for this paper is that the coherence regulation is optimal. However, no empirical evidence against other baseline regularizers are present. Could you provide some comparison with other regularizers?

---

> ### Author Response · Authors · 2026-04-08
>
> > 1. My biggest confusion and concern is with the chain rule assumption. Because maximize $P(\pi^*)$ means the prior should assign high joint probability to the correct answer to every question simultaneously, you would need the chain rule to hold so that the retrained model is a valid instance of prior. If one instead define $D$ to be over the marginals, then coherence optimization seems to reduce to greedy decoding and the results are not interesting anymore. The satisfaction of chain rule is rather counter intuitive for language models, and as the paper correctly pointed out, it seems to be only fully satisfied by the Bayesian learning system (while the other two, arguably, are the learning systems in interest). I wonder if you could provide an approximation error on the chain rule or something, to better quantify how much in context learning system and fine-tuning learning system satisfy this and to better justify coherence optimization?
>
> Thank you for this careful and substantive concern. We want to clarify several points.
>
> First, the chain rule is a modeling assumption in the learning system abstraction. Learning systems provide tractable priors for the SRM framework, and the chain rule ensures that coherence is well-defined (independent of ordering). The theoretical results (Theorems 5.2 and 5.5) hold for any valid learning system. They do not require LLMs to satisfy the chain rule exactly.
>
> Second, we want to address the concern that without the chain rule, coherence reduces to greedy decoding. This is not the case. Even without the exact chain rule, the Gibbs sampling algorithm (Algorithm 1) and the mutual predictability objective (Equation 3, used by ICM) remain well-defined. They optimize conditional predictability across contexts, not marginal likelihood of individual behaviors. The chain rule is needed for the *equivalence* between coherence and joint log-probability, but the practical algorithms optimize conditional predictability directly and do not reduce to greedy decoding.
>
> Third, for ICL systems, the chain rule holds to the extent that ICL approximates Bayesian inference (Xie et al., 2021). The relevant practical concern is order-dependence when shuffling in-context examples. However, coherence optimization only uses well-mixed in-context sequences, so this effect is small. For finetuning systems, the chain rule corresponds to training data order not mattering (again, usually in a well-mixed sequence), which holds to the extent that SGD approximates Bayesian inference (Mingard et al., 2021).
>
> Fourth, a formal bound on chain-rule violation for transformers is an important open question. We have added a Remark ("On the Chain Rule Assumption") in Section 3.2 discussing why the practical algorithms remain meaningful even without exact chain rule satisfaction, and clarifying that they do not reduce to greedy decoding.
>
> > 2. The definition of coherence is on the probability of model assigned to exact outputs. But two words having the same semantics meanings are interpreted as two different exact output. Say if the model have learned $P[yes | yeah] \approx P[yes | yes]$, will coherence treat them similarly? If so, it might be worth to point this out and discuss some of the results under this.
>
> Yes, coherence will treat them similarly. Coherence operates on the probabilities assigned by the inference function $\sigma$, not on string equality. If the prior model has learned that $P[\text{yes} \mid \text{yeah}] \approx P[\text{yes} \mid \text{yes}]$, then "yeah" and "yes" will receive similar conditional probabilities in any given context and thus contribute similarly to coherence computations. Semantic equivalence is handled to the extent that the prior policy treats semantically equivalent outputs similarly, which is a natural property of well-trained language models. We have added a remark after the coherence definition making this point explicit.

---

> ### Author Response · Authors · 2026-04-08
>
> > 3. One big motivation for coherence is for it to be a generalization regularizer. However, Table 1 only reports training accuracy and it seems like no out-of-sample evaluation is performed. Could you add reports on the test accuracy?
>
> Thank you for this observation. We want to make two points.
>
> First, some of our evaluations are already out-of-sample. In the prior policy scaling experiment (Section 6.3), we use LMArena Elo ratings as the ground-truth accuracy measure, which is also an out-of-sample evaluation.
>
> Second, and more fundamentally, practical applications of coherence optimization are typically in-sample by design, similar to transductive learning. The goal is to find the best d-policy over a given set of contexts, where "generalization" goes from the supervised subset to the unsupervised subset of that same context set. Theorem 5.2 bounds accuracy on the full context set, including the unsupervised portion. The unsupervised contexts serve simultaneously as the regularization medium and the evaluation target, which makes train/test splits less relevant than in standard supervised learning. We have added a paragraph in Section 5 discussing this transductive perspective.
>
> That said, we have also run a new experiment with explicit out-of-sample evaluation (Appendix E). We ran 100 seeds per $\gamma$ value using Llama-3.2-1B-Instruct on GSM8K with 100 train questions and 200 held-out test questions. The test questions were never included in the Gibbs sampling context. We report two notions of peak accuracy. The "mean peak" first averages accuracy across seeds at each round, then takes the maximum of this averaged trajectory. The "oracle peak" takes each seed's best round individually, then averages these per-seed maxima (this is higher because each seed selects its own best round). For test accuracy, the mean peak reaches 39-41% across $\gamma$ values, while the oracle peak reaches 46-47%, both up from roughly 30% at initialization. This confirms that coherence optimization generalizes to held-out questions. Interestingly, while in-sample accuracy varies considerably with $\gamma$ (40-45% at the final round), test accuracy is remarkably stable across $\gamma$ values (38-40% final, 39-41% mean peak), suggesting that the generalization benefit is robust to this hyperparameter choice.
>
> > 4. Another important claim for this paper is that the coherence regulation is optimal. However, no empirical evidence against other baseline regularizers are present. Could you provide some comparison with other regularizers?
>
> This is a fair criticism. The paper's contribution is primarily theoretical, proving that coherence is optimal among description-length regularizers. A rigorous empirical comparison would require matching compute budgets across coherence regularization, entropy regularization, KL-divergence to a reference policy, and consistency regularization on a common semi-supervised task. We acknowledge this as an important direction for future work and have expanded the Limitations section to discuss what such a comparison would involve. Section 6.3 provides indirect evidence by showing that coherence-truthfulness alignment improves with prior quality, consistent with the theory.

---

### Review · Reviewer_UEnY · 2026-03-27

**Summary Of Contributions:**

This work focuses on understanding coherence optimization as a unified framework for feedback-free self-improvement in LLMs. This general framework considers the debate, bootstrap, and internal coherence maximization (ICM) methods as special cases. Theoretically, it is shown that coherence optimization is the optimal form of description-length regularization for semi-supervised learning with a pretrained prior.

**Audience:**

Yes

**Audience Explanation:**

The theoretical analysis of coherence optimization provides new insights into unifying existing feedback-free self-improvement approaches.

**Broader Impact Concerns:**

The broader impact has been clearly discussed in the paper.

**Claims And Evidence:**

Yes

**Claims Explanation:**

The examples and detailed proof are provided to support the main claims in the paper.

**Requested Changes:**

(1) It is unclear why "pretrained models provide the best available prior for regularization". How is this justified by Theorem 5.5?

(2) In the abstract, it is highlighted that the theory predicts when feedback-free self-improvement should succeed or fail. This can be explained.

---

> ### Author Response · Authors · 2026-04-08
>
> > (1) It is unclear why "pretrained models provide the best available prior for regularization". How is this justified by Theorem 5.5?
>
> Thank you for raising this. Theorem 5.5 shows that, for description-length regularization with prior $\mathcal{P}$, the worst-case accuracy lower bound is maximized when $\mathcal{P}$ minimizes $\text{KL}[\mathcal{D}^\beta \| \mathcal{P}]$. In other words, the optimal prior is the one closest in KL divergence to the data-generating distribution $\mathcal{D}$. A pretrained language model, trained on the largest and most representative sample of $\mathcal{D}$, is the best practically obtainable such approximation, and therefore yields the tightest generalization bound. We have revised the paragraph following Theorem 5.5 and the corresponding passage in the introduction to make this connection explicit.
>
> > (2) In the abstract, it is highlighted that the theory predicts when feedback-free self-improvement should succeed or fail. This can be explained.
>
> We agree this deserved more explanation. The theory predicts success and failure through two results. First, Theorem 5.5 shows that the accuracy lower bound is maximized when the prior minimizes KL divergence to the data-generating distribution. When the prior is well-aligned, the bound is tight and self-improvement succeeds. When the prior is misaligned, the bound becomes vacuous and self-improvement fails. Second, Conjecture 5.9 characterizes when optimizing coherence alone (without an explicit accuracy term) recovers the full regularized training objective. The equivalence holds when the number of unsupervised questions is chosen appropriately relative to the effective number of supervised samples. When this balance is violated, coherence optimization can fail. We have added a footnote in the abstract pointing to these two results.

---

### Decision · Action_Editor_NCLB · 2026-05-27

**Recommendation:** Accept with minor revision

**Additional Comments:**

The reviewers are overall positive about the contributions:
- "The paper provide a general framework to understand several method that helps improve language model accuracy without external supervision." (Reviewer LjMB)

- "The learning system is well set up and the theoretical results are well written." (Reviewer LjMB)

- "provide theoretical analysis of recent empirical advances in self-improving large models. This is quite rare and represents a strong positive contribution." (Reviewer BNBn)

- "Efforts are made to connect the theory to practical applications." (Reviewer BNBn)

The reviewers also voiced some concerns:
- "Another important claim for this paper is that the coherence regulation is optimal. However, no empirical evidence against other baseline regularizers are present" (Reviewer LjMB), as well as "Preliminary experiment results do show some advantage for their proposed algorithm, but are less convincing due to the very few benchmarks and lack of realistic baselines." (Reviewer BNBn).

- "Some parts of the claim that coherence optimization is an umbrella unifying framework for existing techniques are a bit weak" (Reviewer BNBn), which I agree since both simple bootstrap and internal coherence maximization do not fall as special cases of the proposed framework; at best they share some similarities. It is advised that the authors tune this claim down a bit.

There are some other issues that I believe  worth addressing:
- Coherence optimization is never formally defined. Presumably it means maximizing $\pi$ wrt to $\chi_{\phi}(\pi)$ in Definition 3.6. If this is the case, the authors need to explicitly discuss
  - why maximizing $\pi$ directly is difficult.
  - how the proposed algorithm sidesteps the difficulty, in particular, how Definition 3.7 is relevant and what role does $\beta$ play.
  - my understanding is that one turns the maximization problem into sampling from a Gibbs distribution, which is a very standard trick in Bayesian analysis (i.e., maximum a posteriori vs sampling from the posterior). There is a large body of works on this (in stats and ML, e.g., early works on the Boltzmann machine), but the authors did not cite any of them! Even the original work of Geman and Geman on Gibbs sampling was not cited, which may give the wrong impression that the presented Gibbs sampling algorithm or Theorem 4.2 is new. (I believe this is mostly a writing issue than intentional.) Please address this.
   - moreover, Gibbs sampling does not maximize coherence per se; the authors need to discuss the role of $\beta$ (as in simulated annealing) and how fast the underlying chain is mixing. Just because one can run Gibbs sampling does not mean we are maximizing the coherence *tractably*. What do we mean by "tractable" anyway?

- in all the results, $-log(1/\delta)$ should be $log(1/\delta)$?

- the authors seem to draw optimality of coherence maximization from maximizing a lower bound of the agreement (Theorem 5.5). This is not uncommon but I'd caution against such claims: we can always subtract a nonnegative regularization term from a lower bound and claim "optimality" of this regularization by maximizing the resulting lower bound. Even when the lower bound is tight, claim on optimality is weak: it only means in the worst-case (when the lower bound is attained), things cannot get better, but in practice one may rarely be in the worst-case and it may still be possible to beat the so-claimed "optimal" method. Regrettably, there are not enough comparisons against other baselines to support the optimality claim.

- the abstract also claimed that "it is optimal for semi-supervised learning when the regularizer is derived from a pretrained model." It is not clear which part of the paper formally proved this claim. In particular, it is not clear how Conjecture 5.9 in Section 5.3 affects this claim. This subsection feels a bit semi-finished.

**Audience:**

Yes

**Audience Explanation:**

This work provides a formalization to facilitate discussions on self-improving LLMs. Its framework encompasses some recent related works and builds connections to other fields (Gibbs sampling and philosophical epistemology).

**Claims And Evidence:**

Yes

**Claims Explanation:**

Most of the claims are explained through examples, discussions, proofs and some preliminary experimental verifications.